# Rab10 regulates neuropeptide release by maintaining Ca²⁺ homeostasis and protein synthesis

Jian Dong[1], Miao Chen[2], Jan RT van Weering[3], Natalia Domínguez[1,3†], Ka Wan Li[2], August B Smit[2], Ruud F Toonen[1], Matthijs Verhage[1,3*]

[1]Department of Functional Genomics, Center for Neurogenomics and Cognitive Research (CNCR), Vrije Universiteit (VU) Amsterdam, Amsterdam, Netherlands; [2]Department of Molecular and Cellular Neurobiology, Center for Neurogenomics and Cognitive Research (CNCR), Vrije Universiteit (VU) Amsterdam, Amsterdam, Netherlands; [3]Department of Clinical Genetics, Center for Neurogenomics and Cognitive Research (CNCR), University Medical Center Amsterdam, Amsterdam, Netherlands

*For correspondence: matthijs@cncr.vu.nl

Present address: †Departamento de Bioquímica, Microbiología, Biología Celular y Genética, Universidad de La Laguna, La Laguna, Spain

## eLife Assessment

In this revised manuscript, Dong et al. investigate the role of the small Ras-like GTPase Rab10 in the exocytosis of DCVs in mouse hippocampal neurons, showing that Rab10 depletion hinders DCV exocytosis independently of its effects on neurite outgrowth. Upon revising their work, these findings provide **compelling** evidence that Rab10 depletion leads to altered ER morphology, impaired ER-based calcium buffering, and decreased ribosomal protein expression, which collectively contributes to defective DCV secretion. The study comes to the **fundamental** conclusion that Rab10 is critical for DCV release by ensuring ER calcium homeostasis.

**Abstract** Dense core vesicles (DCVs) transport and release various neuropeptides and neurotrophins that control diverse brain functions, but the DCV secretory pathway remains poorly understood. Here, we tested a prediction emerging from invertebrate studies about the crucial role of the intracellular trafficking GTPase Rab10, by assessing DCV exocytosis at single-cell resolution upon acute Rab10 depletion in mature mouse hippocampal neurons, to circumvent potential confounding effects of Rab10's established role in neurite outgrowth. We observed a significant inhibition of DCV exocytosis in Rab10-depleted neurons, whereas synaptic vesicle exocytosis was unaffected. However, rather than a direct involvement in DCV trafficking, this effect was attributed to two ER-dependent processes, ER-regulated intracellular Ca²⁺ dynamics, and protein synthesis. Gene Ontology analysis of differentially expressed proteins upon Rab10 depletion identified substantial alterations in synaptic and ER/ribosomal proteins, including the Ca²⁺ pump SERCA2. In addition, ER morphology and dynamics were altered, ER Ca²⁺ levels were depleted, and Ca²⁺ homeostasis was impaired in Rab10-depleted neurons. However, Ca²⁺ entry using a Ca²⁺ ionophore still triggered less DCV exocytosis. Instead, leucine supplementation, which enhances protein synthesis, largely rescued DCV exocytosis deficiency. We conclude that Rab10 is required for neuropeptide release by maintaining Ca²⁺ dynamics and regulating protein synthesis. Furthermore, DCV exocytosis appeared more dependent on (acute) protein synthesis than synaptic vesicle exocytosis.

## Introduction

Dense core vesicles (DCVs) transport and release neuromodulators (neuropeptides, neurotrophic factors, and catecholamines) that play crucial roles in modulating diverse brain functions, including sleep, mood, memory, and learning (e.g. *Cawley et al., 2016*; *Malva et al., 2012*; *Poo, 2001*; *Salio et al., 2006*). Deficits in neuropeptide signaling pathways have been associated with several human disorders and diseases, including anxiety, depression, and obesity (*Beck, 2000*; *Sah and Geracioti, 2013*; *Barde et al., 2022*). Neuropeptides are synthesized in the endoplasmic reticulum (ER) and subsequently packaged into immature DCVs in the Golgi complex (*Tooze et al., 2001*). DCVs are transported along the neurites and undergo activity-dependent membrane fusion with the plasma membrane to release their content (*Farina et al., 2015*; *Heidelberger et al., 1994*; *Nassal et al., 2022*; *Thomas et al., 1993*; *van de Bospoort et al., 2012*). Despite the critical role of neuropeptides in brain functions, the regulatory mechanisms governing their secretion are not fully understood. We have shown previously that Rab3 is an essential regulator in the last steps of the DCV secretory pathway in mammalian neurons (*Persoon et al., 2019*). However, studies in invertebrates have also implicated other Rab proteins, including Rab2, Rab5, and Rab10, in the DCV secretory pathway (*Ailion et al., 2014*; *Azouz et al., 2014*; *Edwards et al., 2009*; *Hannemann et al., 2012*; *Lund et al., 2020*; *Sasidharan et al., 2012*; *Sumakovic et al., 2009*).

Among these Rab proteins, Rab10 deficiency produces the strongest inhibition of neuropeptide release in *Caenorhabditis elegans* (*Sasidharan et al., 2012*). With its subcellular localization in many vesicular organelles, such as plasmalemmal precursor vesicles, GLUT4 transport vesicles, and recycling endosomes, Rab10 regulates various aspects of intracellular membrane trafficking, including vesicle formation, transport, and fusion (*Chen et al., 2006*; *Larance et al., 2005*; *Mîinea et al., 2005*; *Sano et al., 2007*; *Taylor et al., 2015*). Rab10 deficiency leads to deficits in these pathways, resulting in impaired neuronal outgrowth and disrupted retrograde axonal transport of signaling factors (*Lazo and Schiavo, 2023*). How these deficits relate to the strong inhibition of DCV exocytosis remains unknown.

Deletion of Rab10 expression or inhibiting its functions by overexpression of an inactive mutant (Rab10T23N) leads to abnormal ER morphology (*English and Voeltz, 2013*; *Lv et al., 2015*; *Shih and Hsueh, 2016*). Since the ER is a crucial organelle involved in protein synthesis, $Ca^{2+}$ buffering, and lipid metabolism, alterations in its structure and function can directly or indirectly affect neuronal secretory pathways. Indeed, several studies have suggested the roles of the ER in the DCV pathway, such as the involvement of ER stress and lipid levels in DCV production in *C. elegans* (*Laurent et al., 2018*; *Valadas et al., 2018*), and the roles of ER $Ca^{2+}$ as an internal $Ca^{2+}$ source regulating somatodendritic dopamine release in mouse substantia nigra neurons (*Patel et al., 2009*). Additionally, Rab10 mutations, altered expression levels, or phosphorylation states are firmly associated with CNS disorders (*Agola et al., 2011*; *Cheng et al., 2005*; *Kiral et al., 2018*). Hence, the strong inhibition of neuropeptide accumulation in coelomocytes in nematode Rab10 mutants may be a direct or indirect effect of Rab10 loss of function on DCV exocytosis and Rab10-dependent disease processes may (in part) be explained by DCV exocytosis impairment.

In the present study, we investigated the involvement of Rab10 in regulated secretion of neuropeptides in mammalian neurons. We directly assessed DCV exocytosis at single-cell resolution in mouse hippocampal neurons and confirmed that Rab10 is a crucial regulator of DCV exocytosis also in mammalian systems, while not affecting synaptic vesicle (SV) exocytosis. However, instead of having a direct role in the DCV secretory pathway, we observed that Rab10 is involved in DCV exocytosis through ER-dependent processes, especially reduced ER $Ca^{2+}$ homeostasis and impaired global protein synthesis. Supplementation with leucine known to boost protein synthesis rescued the defects in DCV exocytosis. We therefore conclude that Rab10 plays a central role in regulating DCV exocytosis by maintaining $Ca^{2+}$ homeostasis and protein synthesis.

## Results

### Rab10 regulates neuronal outgrowth but is dispensable for synaptogenesis and SV exocytosis under intense stimulation

To study the role of Rab10 in regulated secretion, we utilized a knockdown strategy since complete knockout of Rab10 results in lethality at cell and organismal levels (*Lv et al., 2015*). We selected

two specific shRNA sequences, shRNA#9 and shRNA#11, to deplete Rab10 expression and a scrambled sequence as control. In mouse cortical neurons infected with either shRNA#9 or shRNA#11 at day in vitro 0 (DIV0), we observed a 75–95% decrease in Rab10 expression at DIV14 (*Figure 1A*). Previous studies have shown that Rab10 regulates neuronal outgrowth (*Wang et al., 2011*; *Xu et al., 2014*). Consistent with these findings, we observed a significantly reduced dendrite and axon length in neurons infected with shRNA#9 at DIV0 (*Figure 1*).

To test the effects of Rab10 depletion on synaptogenesis, SVs were quantified using the endogenous marker synaptophysin 1 (Syp1). Syp1 staining exhibited a punctate distribution at DIV14, indicating the accumulation of SVs in boutons/synapses, and no changes in the number of puncta per µm neurite or the intensity of Syp1 puncta (*Figure 1E and F*). These data confirm that Rab10 regulates neurite outgrowth, but we found no evidence for a role in synaptogenesis.

To test whether Rab10 depletion affects SV exocytosis, hippocampal neurons were infected with the SV exocytosis reporter Synaptophysin-pHluorin (SypHy; *Figure 1H*; *Granseth et al., 2006*). SV exocytosis was triggered by high-frequency electrical stimulation (HFS, 5 s 40 Hz). The total vesicle pool was measured by briefly superfusing Tyrode's solution containing 50 mM $NH_4Cl$. The fraction of fused SVs, determined by the ratio of SypHy intensity upon HFS to the maximum intensity upon $NH_4Cl$ superfusion, was comparable in the two groups (*Figure 1I and J*). In addition, SV endocytosis, measured by the fluorescence decay of SypHy after HFS, was unaffected by Rab10 depletion (*Figure 1K*). Therefore, we conclude that Rab10 is dispensable for SV exocytosis under intense stimulation.

## Rab10 is a major regulator of DCV exocytosis

To investigate the role of Rab10 in neuropeptide release, we depleted Rab10 levels at DIV0, expressed the DCV exocytosis reporter NPY-pHluorin at DIV9-10, and performed live-cell imaging at DIV14. Our previous studies have demonstrated that NPY-pHluorin almost exclusively localizes to DCVs as indicated by its strong co-localization with endogenous markers of DCVs, such as BDNF and the chromogranins CHGA and CHGB (*Arora et al., 2017*; *Dominguez et al., 2018*; *Persoon et al., 2019*; *Persoon et al., 2018*). To achieve single-vesicle resolution analysis of DCV exocytosis, we used single cultured hippocampal neurons on glial micro-islands. Neurons were stimulated by 16 trains of 50 action potentials (APs) at 50 Hz, a protocol known to trigger robust DCV exocytosis (*Balkowiec and Katz, 2002*; *Emperador-Melero et al., 2018*; *Gärtner and Staiger, 2002*; *Hartmann et al., 2001*; *Moro et al., 2021*; *Persoon et al., 2019*). Fusion events were detected as a rapid appearance of fluorescent puncta (*Figure 2—figure supplement 1A and B*). DCV exocytosis in Rab10 KD neurons was significantly reduced by 60% compared to control neurons (*Figure 2—figure supplement 1C and D*). Furthermore, the total number of DCVs was reduced by 30% in Rab10 KD neurons (*Figure 2—figure supplement 1E*). The fusion fraction, which represents the number of DCV fusion events relative to the remaining DCV pool after stimulation, was also significantly decreased by 50% in Rab10 KD neurons (*Figure 2—figure supplement 1F*). Overexpression of wild-type (WT), knockdown-resistant Rab10 rescued DCV exocytosis deficits in Rab10 KD neurons (*Figure 2—figure supplement 1C–F*).

To overcome the potential confounding effects of impaired neurite outgrowth (*Figure 1*) and reduced total DCV pool (*Figure 2—figure supplement 1*) upon Rab10 depletion starting at DIV0, we adopted a more acute approach to interfere with Rab10 function, and late enough not to affect neuronal morphology and the total DCV pool. Neurons were transfected with shRNA against Rab10 at DIV7, fixed at DIV14, and stained with markers for dendrites (MAP2), axons (SMI312), and SVs (Syp1). Rab10 expression was reduced by 70% after 7 days of infection (*Figure 3—source data 1*). No significant alterations in total dendrite length, axon length (*Figure 2A–C and E*), or synapse density (*Figure 2D*) were observed in Rab10 KD neurons under these conditions. Therefore, to eliminate confounding effects on morphological parameters, we reevaluated DCV exocytosis and all further experiments in neurons infected with shRNA at DIV7. DCV exocytosis in Rab10 KD neurons remained significantly lower by 50% compared to control neurons (*Figure 2F–I*). The fusion fraction was also significantly reduced by 65% in Rab10 KD neurons (*Figure 2K*). Overexpression of WT Rab10 rescued DCV exocytosis deficits. No significant differences in the total number of DCVs (*Figure 2J*), DCV transport (*Figure 2—figure supplement 2A, B, and E*), or cargo loading (*Figure 2—figure supplement 2F–K*) were observed. Moreover, only 10% of DCVs co-transport with Rab10 (*Figure 2—figure supplement 3*). Thus, these data indicate that Rab10 depletion specifically inhibits activity-dependent neuropeptide release in hippocampal neurons, without effects on DCV

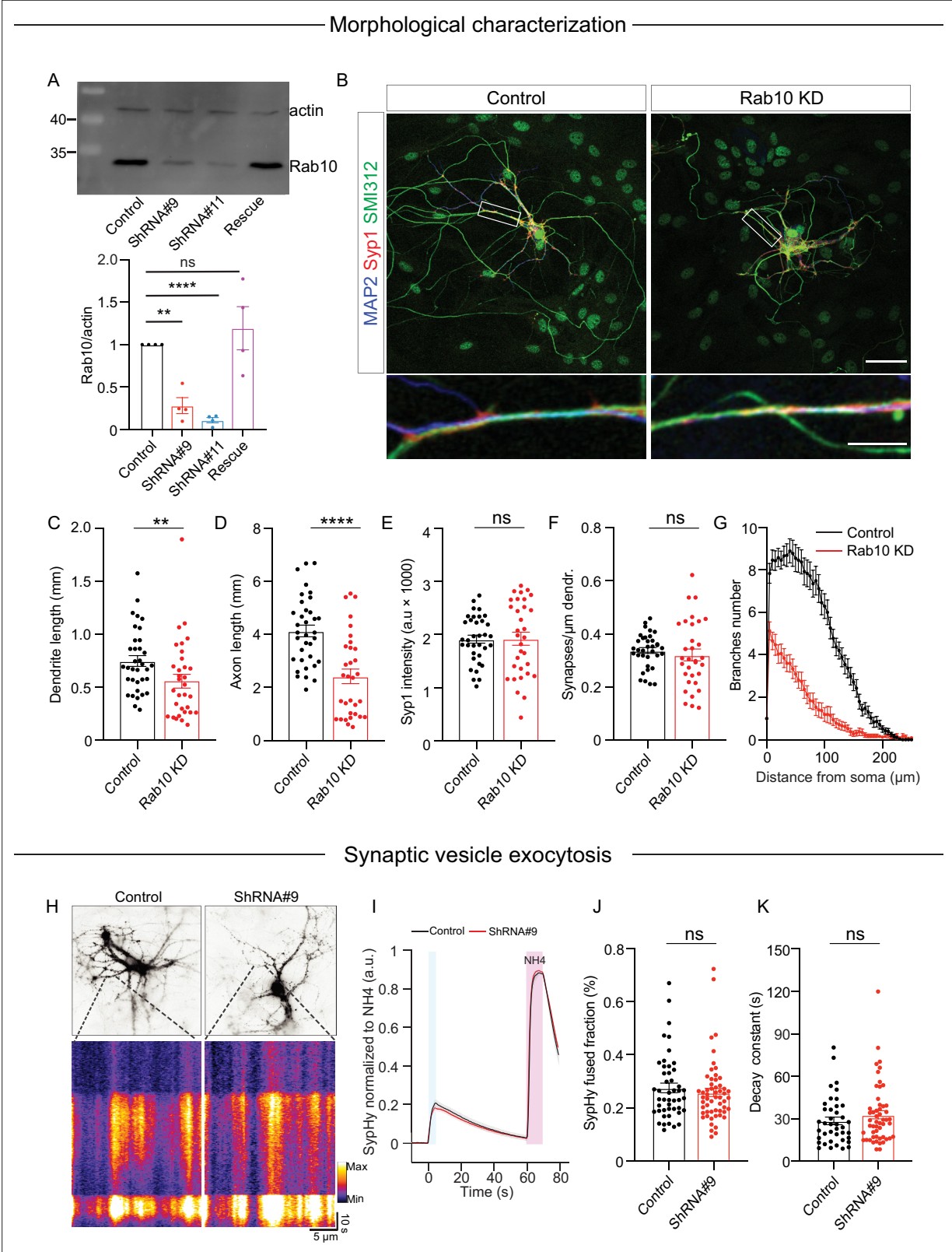

**Figure 1.** Rab10 is required for neuronal outgrowth but dispensable for synaptic vesicle (SV) exocytosis evoked by intense stimulations.
(**A**) Representative immunoblotting showing knockdown and rescue of Rab10 expression in cultured primary neurons infected with shRNA against Rab10 or rescue constructs (upper) and quantification of Rab10 levels (bottom). (**B**) Example images of control or Rab10 KD hippocampal neurons (days in vitro [DIV]14) stained for the dendrite marker MAP2 (blue), the synapse marker Syp1 (red), and the axonal marker SMI312 (green). Scale bar: 50 µm

*Figure 1 continued on next page*

*Figure 1 continued*

(upper) and 10 µm (bottom). (**C**) Quantification of the dendritic length (MAP2). (**D**) Quantification of the axonal length (SMI312). (**E**) Quantification of Syp1 intensity per synapse per neuron. (**F**) Quantification of the Syp1-positive synapse density in MAP2-positive dendrites. (**G**) Sholl analysis showing the mean number of dendritic branches against the distance from the soma. (**H**) Example neurons infected with the SV fusion marker SypHy (upper), typical kymographs of neurites showing SypHy intensity increase during stimulation and upon NH₄Cl superfusion (bottom). (**I**) The average signal SypHy from active synapses, normalized from baseline to maximum fluorescence upon NH₄Cl superfusion. (**J**) SV exocytosis determined as the ratio of the maximum SypHy intensity during stimulation to the maximum during NH₄Cl stimulation. (**K**) SV endocytosis determined as the SypHy signal decay time constant $\tau$ in the 60 s after field stimulation. All data are plotted as mean ± s.e.m. (**A**) N=4, n=4, one-sample t-test. (**C–G**) Control: N=3, n=35; ShRNA#9: N=3, n=32. (**J, K**) Control: N=3, n=47; ShRNA#9: N=3, n=56. (**C–F, J, K**) A one-way ANOVA tested the significance of adding experimental group as a predictor. ****=p<0.0001, **=p<0.01, ns=not significant.

The online version of this article includes the following source data for figure 1:

**Source data 1.** PDF file containing original western blots for *Figure 1A*, indicating the relevant bands and treatments.

**Source data 2.** Original files for western blot analysis displayed in *Figure 1A*.

biogenesis, cargo loading, and transport and independent of Rab10's established role in neuronal outgrowth.

## Proteins involved in synaptic transmission and translation are severely dysregulated upon Rab10 depletion

To comprehensively investigate Rab10 function in mature neurons, mass spectrometry proteomics was performed on Rab10 KD and control neurons at DIV14. A total of approximately 5400 unique proteins were identified and quantified in two biological replicates. The complete list of proteins quantified in this study is presented in *Figure 3—source data 1*. Only differentially expressed proteins detected with high confidence characterized by a log2(fold change)>0.56 and q-value<0.01 were included in the subsequent analysis. Among the dysregulated proteins, 71% were upregulated, while 29% were downregulated, resulting in a significant dysregulation of 19% of the total protein pool in Rab10 KD neurons. These data indicate that Rab10 depletion leads to major neuronal proteome dysregulation within 7 days after initiating knockdown (*Figure 3A*).

To gain insights into the functional consequences of Rab10 depletion, we performed Gene Ontology (GO) analysis using ClueGO (*Bindea et al., 2009*). This analysis revealed that biological processes related to chemical synaptic transmission were notably affected by Rab10 depletion (*Figure 3B*). In addition, several biological processes related to protein synthesis, such as cytoplasmic translation and ribosomal large subunit biogenesis, were among the top 5 terms with the lowest p-values (*Figure 3B*). Subcellular localization analysis of these dysregulated proteins upon Rab10 depletion showed that cytosolic ribosomal proteins were the most significantly affected, followed by dendritic proteins (*Figure 3C*). Further characterization of the dysregulated synaptic proteins was performed using SynGO (*Koopmans et al., 2019*). Among the 391 significantly dysregulated proteins annotated in SynGO, 205 were classified as presynaptic proteins and 237 as postsynaptic proteins. GO enrichment analysis revealed that biological processes in metabolism were most dysregulated upon Rab10 depletion. Among them, both pre- and postsynaptic translation were significantly dysregulated (*Figure 3D*). Consistent with the ClueGO analysis, SynGO highlighted a significant dysregulation of presynaptic (34 of 391) and postsynaptic ribosomal proteins (44 of 391), supporting the involvement of Rab10 in the regulation of neuronal protein synthesis (*Figure 3E*). Interestingly, all dysregulated ribosomal proteins were upregulated upon Rab10 depletion, which contrasts with the mostly downregulated expression observed in the other classes of proteins (such as synaptic and cytoskeletal proteins) (*Figure 3F–G*). Taken together, GO analysis with both ClueGO and SynGO indicates a dysfunction of protein translation in Rab10 KD neurons.

Loss of Rab10 expression has been associated with altered ER morphology in mouse embryonic cells (*Lv et al., 2015*), which may explain the selective upregulation of proteins involved in ribosome function in our proteomics data. Indeed, ER proteins were dysregulated substantially upon Rab10 depletion. Specifically, several rough ER (RER) proteins showed differential regulation, with SEC61A1 being upregulated, SEC61G being downregulated, and CLIMP remaining unchanged (*Figure 3H*). Most tubular ER proteins, such as RTN3/4 and VAPB, were robustly decreased. Interestingly, one of the ER membrane $Ca^{2+}$ channels, SERCA2, showed a 50% reduction upon Rab10 depletion (*Figure 3H*).

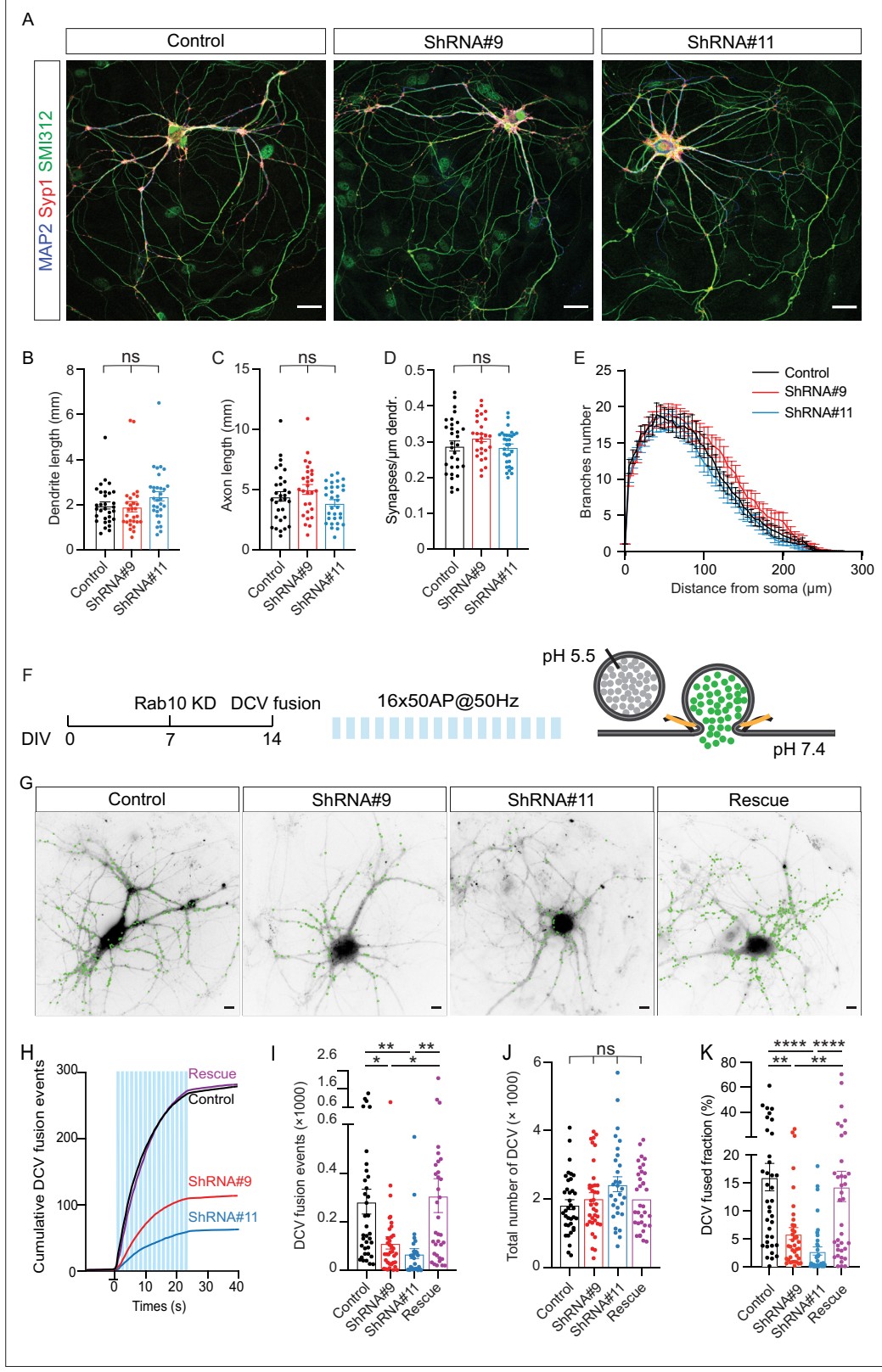

**Figure 2.** Rab10 is a major regulator of dense core vesicle (DCV) exocytosis. (**A**) Example images of control and Rab10 KD hippocampal neurons (days in vitro [DIV]14) stained for MAP2 (blue), Syp1 (red), and SMI312 (green). Scale bar: 30 µm. (**B**) Quantification of the dendritic length (MAP2). (**C**) Quantification of the axonal length (SMI312). (**D**) Quantification of the Syp1-positive synapse density in MAP2-positive dendrites. (**E**) Sholl analysis

*Figure 2 continued on next page*

*Figure 2 continued*

showing the mean number of dendritic branches against the distance from the soma. (**F**) Schematic representation of DCV fusion assay. DCVs are labeled with NPY-pHluorin, and neurons are stimulated with one train of 16 bursts of 50 action potentials (APs) at 50 Hz (light blue bars). (**G**) Representative neurons during electrical stimulation superimposed with NPY-pHluorin fusion events (green dots). Scale bar: 5 µm. (**H**) Cumulative plot of DCV fusion events per cell. Light blue bars represent the stimulation trains. (**I**) Summary graph of DCV fusion events per cell. (**J**) The total number of DCVs (total pool) of neurons analyzed in H, measured as the number of NPY-pHluorin puncta upon $NH_4Cl$ perfusion. (**K**) Fraction of NPY-pHluorin-labeled DCV fusing during stimulation. All data are plotted as mean ± s.e.m. (**B–D**) Control: N=3, n=31; ShRNA#9: N=3, N=28; ShRNA#11: N=3, n=31. (**I–K**) Control: N=4, n=36; shRNA#9: N=4, n=37; shRNA#11: N=4, n=30; Rescue: N=4, n=34. A one-way ANOVA tested the significance of adding experimental group as a predictor. ****=p<0.0001, ***=p<0.001, **=p<0.01, *=p<0.05, ns=not significant.

The online version of this article includes the following figure supplement(s) for figure 2:

**Figure supplement 1.** Rab10 depletion at day in vitro (DIV)0 impedes dense core vesicle (DCV) fusion.

**Figure supplement 2.** Rab10 depletion does not affect dense core vesicle (DCV) transport or cargo loading.

**Figure supplement 3.** Rab10 does not typically co-transport together with dense core vesicles (DCVs).

Taken together, these analyses reveal that Rab10 depletion leads to major changes in protein expression, especially synaptic and ribosomal proteins, the latter all upregulated in Rab10-depleted neurons which suggests that protein synthesis is dysregulated, potentially due to altered ER function.

## Rab10 regulates ER morphology and ribosomal protein levels

Given the substantial dysregulation of synapse and ribosome/ER proteins, we investigated synapses and ER further using electron microscopy (*Figure 4*). These analyses revealed an apparently normal synapse morphology in Rab10-depleted neurons with many SVs clustered at the active zone, while DCVs were sparsely distributed along neurites and near the active zone (*Figure 4A*). The length of the active zone and postsynaptic density were both decreased by 10% upon Rab10 depletion (*Figure 4C and D*). However, other parameters of synaptic ultrastructure, such as the diameter of SVs or DCVs, and the number of SVs per synapse, remain unchanged in Rab10 KD neurons (*Figure 4E–H*). Hence, despite substantial dysregulation of synaptic proteins, overall synapse morphology was hardly affected.

Tubular ER was also observed in some presynaptic sections, consistent with previous studies (*Deng et al., 2021*; *Droz et al., 1975*; *Wu et al., 2017*). The percentage of synaptic sections containing tubular ER was decreased by 24% (37% in control versus 28% in Rab10 KD). rER was also studied in the somata of Rab10-depleted and control neurons (*Figure 4I*). The diameter of rER tubes was reduced by 15% in Rab10 KD neurons. Hence, the substantial dysregulation of ribosomal and ER proteins in Rab10-depleted neurons was accompanied by changes in the abundance of synaptic ER and small changes in ER morphology in the soma.

To study these effects on ER abundance and morphology further, we performed immunofluorescence staining for two endogenous ER markers, KDEL and RTN4. The average fluorescence intensity of RTN4 and KDEL staining was significantly decreased by 35% and 25% respectively in Rab10 KD neurons (*Figure 4—figure supplement 1A–C*). The relative distributions of RTN4 and KDEL in neurites, as calculated by the intensity ratio of these two proteins in neurites over their somatic intensity, were reduced by 25% and 13%, respectively (*Figure 4—figure supplement 1D and E*). In conclusion, the ultrastructural changes in ER abundance and morphology upon Rab10 depletion were accompanied by altered distribution of axonal ER, without affecting the ultrastructure of SVs and DCVs.

Finally, given the dynamic nature of ER tubular networks and the involvement of Rab10 in ER tubule extension in COS-7 cells (*English and Voeltz, 2013*), we tested ER dynamics in Rab10 KD and control neurons expressing the luminal ER marker mCherry-ER3 using live-cell imaging at DIV8, and performed FRAP experiments at DIV14. The recovery of mCherry-ER3 intensity after photobleaching was significantly slower in Rab10 KD neurons with only 50% recovery within 3 min, compared to 80% recovery in control neurons (*Figure 4—figure supplement 2C*). Collectively, these data indicate that Rab10 depletion leads to reduced levels of ER-resident proteins altered ER abundance and morphology, and impaired ER dynamics.

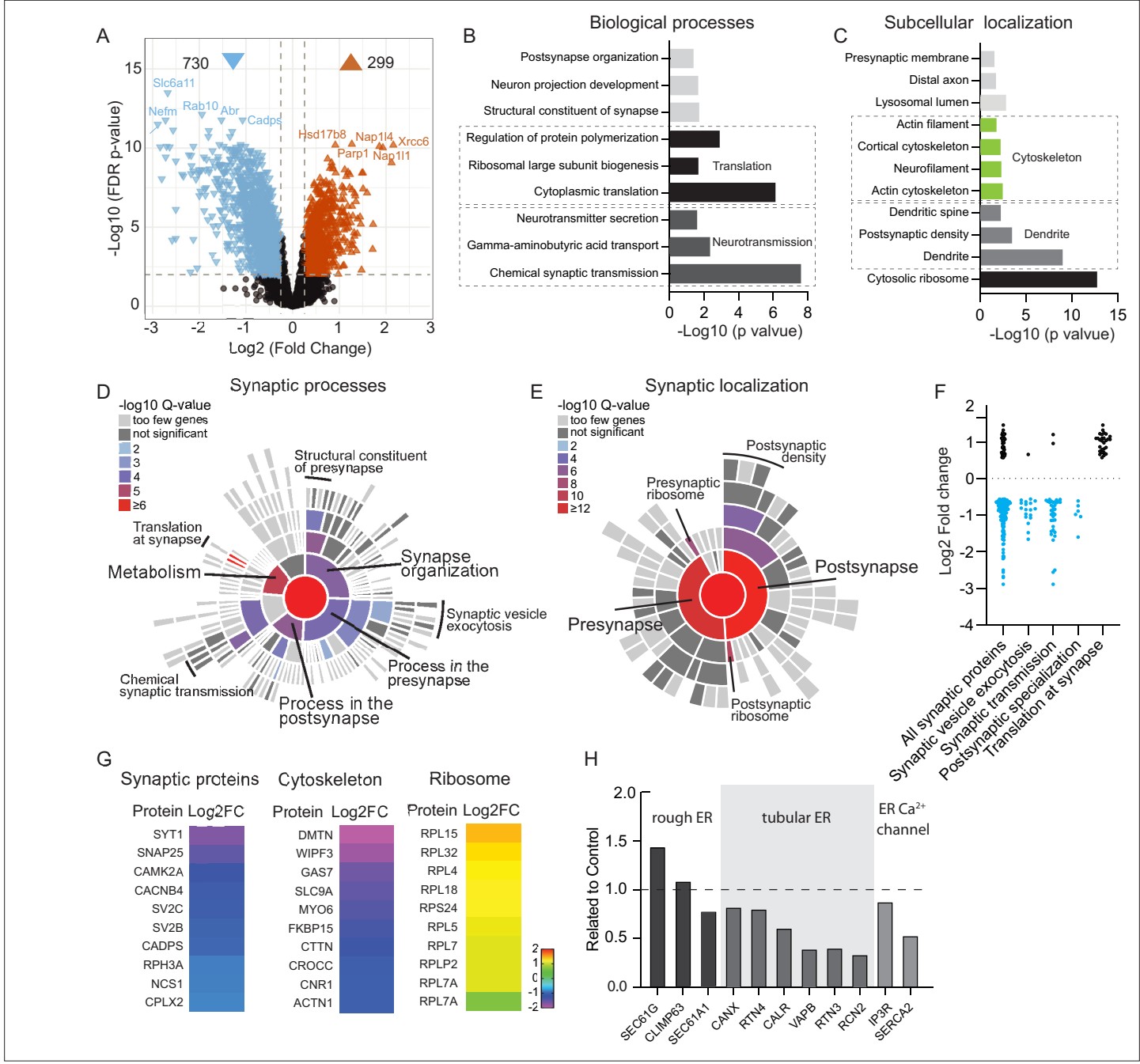

**Figure 3.** Depletion of Rab10 leads to dysregulation of proteins enriched in presynaptic transmission and cytosolic translation. (**A**) Volcano plots showing significantly dysregulated proteins in Rab10-depleted neurons. (**B**) Gene Ontology (GO) enrichment analysis of functional pathways of the significant hits with ClueGO. Shown are the Bonferroni corrected p-values. (**C**) GO enrichment analysis of subcellular localization of the significant hits with ClueGO. Shown are the Bonferroni corrected p-values. (**D**) Sunburst plot showing the annotation in synaptic function of the altered proteins in Rab10-depleted neurons. (**E**) Sunburst plot showing the annotation in synaptic location of the altered proteins in Rab10-depleted neurons. (**F**) Log2 fold changes of synaptic proteins within SynGO terms. Downregulated proteins are shown in blue and upregulated proteins are shown in black. (**G**) Examples of proteins that are significantly affected by Rab10 depletion grouped by their subcellular localization. Heat maps represent the degree of up- or downregulation. (**H**) Selective MS data analysis of ER-related proteins in Rab10 KD neurons. Bars show the fold change of the indicated peptides compared to the control.

The online version of this article includes the following source data for figure 3:

**Source data 1.** Proteome analysis of neuronal cultures by mass spectrometry – complete list of proteins.

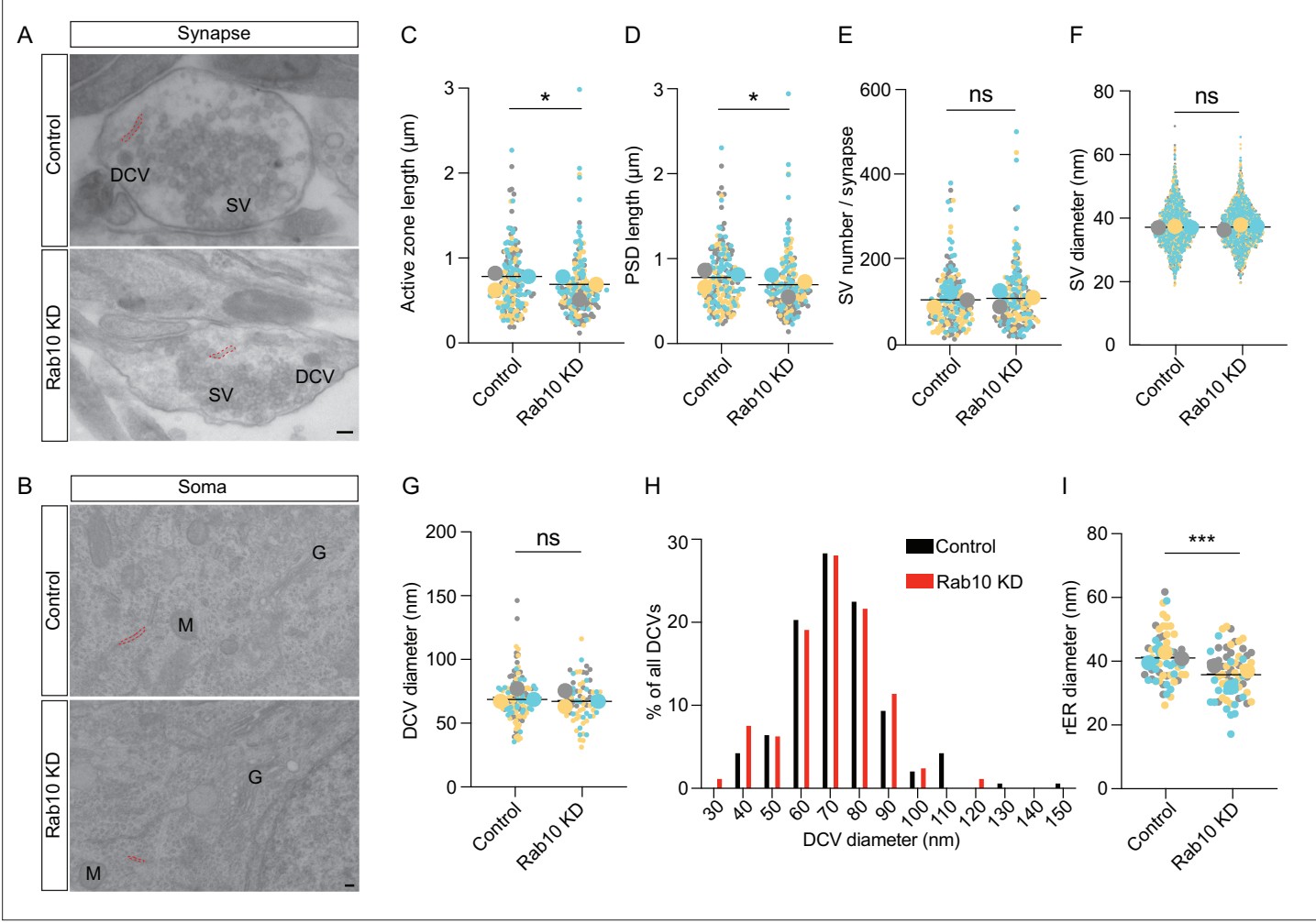

**Figure 4.** Rab10 regulates synapse size and endoplasmic reticulum (ER) morphology. (**A**) Representative electron microscopy (EM) pictures showing the ultrastructure of synapses. Scale bar: 100 nm. Synaptic ER is indicated by red dotted lines. (**B**) Representative EM pictures showing the ultrastructure of soma. Rough ER (rER) is indicated by red dotted lines. M: mitochondrion, G: Golgi. Scale bar: 100 nm. (C) Quantification of the length of active zone and postsynaptic density (PSD). (D) Quantification of the length of PSD. (E) Quantification of synaptic vesicle (SV) number per synapse and SV diameter. (F) Quantification of SV diameter. (G) Quantification of dense core vesicle (DCV) diameter. (H) Frequency distribution of DCVs by diameter. (I) Quantification of the diameter of rER. Data are plotted with superplot (C–G, I), where averages from three independent cultures are shown as large circles and single observations are shown as dots. Horizontal lines represent the means of the averages from 3 weeks. Data from different cultures are grouped with different colors. (**C–D**) Control: N=3, n=184; shRNA#9: N=3, n=187. (**E**) Control: N=3, n=189; shRNA#9: N=3, n=188. (**F**) Control: N=3, n=1770; shRNA#9: N=3, n=1803. (**G**) Control: N=3, n=137; shRNA#9: N=3, n=122. (**I**) Control: N=3, n=63; shRNA#9: N=3, n=64. (**C–G, I**) Linear mixed model analysis. ***=p<0.001, *=p<0.05, ns=not significant.

The online version of this article includes the following figure supplement(s) for figure 4:

**Figure supplement 1.** Altered endoplasmic reticulum (ER) morphology in Rab10 KD neurons.

**Figure supplement 2.** Impaired endoplasmic reticulum (ER) dynamics in Rab10 KD neurons.

**Figure supplement 3.** Rab10 depletion does not increase endoplasmic reticulum (ER) stress.

## Rab10 regulates SERCA2 levels and ER Ca²⁺ homeostasis

The ER is the largest internal $Ca^{2+}$ source in neurons and plays a crucial role in maintaining neuronal $Ca^{2+}$ homeostasis (*Karagas and Venkatachalam, 2019*). The maintenance of ER $Ca^{2+}$ primarily relies on the Sarco Endoplasmic Reticulum Calcium ATPase (SERCA), a $Ca^{2+}$ pump. Among the three major paralogs of SERCA, SERCA2 is particularly enriched in neurons (*Britzolaki et al., 2018*; *Periasamy and Kalyanasundaram, 2007*; *Xu and Van Remmen, 2021*). Consistent with the proteomic analysis which revealed a reduced SERCA2 expression upon Rab10 depletion (*Figure 3*), immunoblotting confirmed the reduction of SERCA2 levels, showing a 50% reduction in Rab10 KD neurons (*Figure 5A*

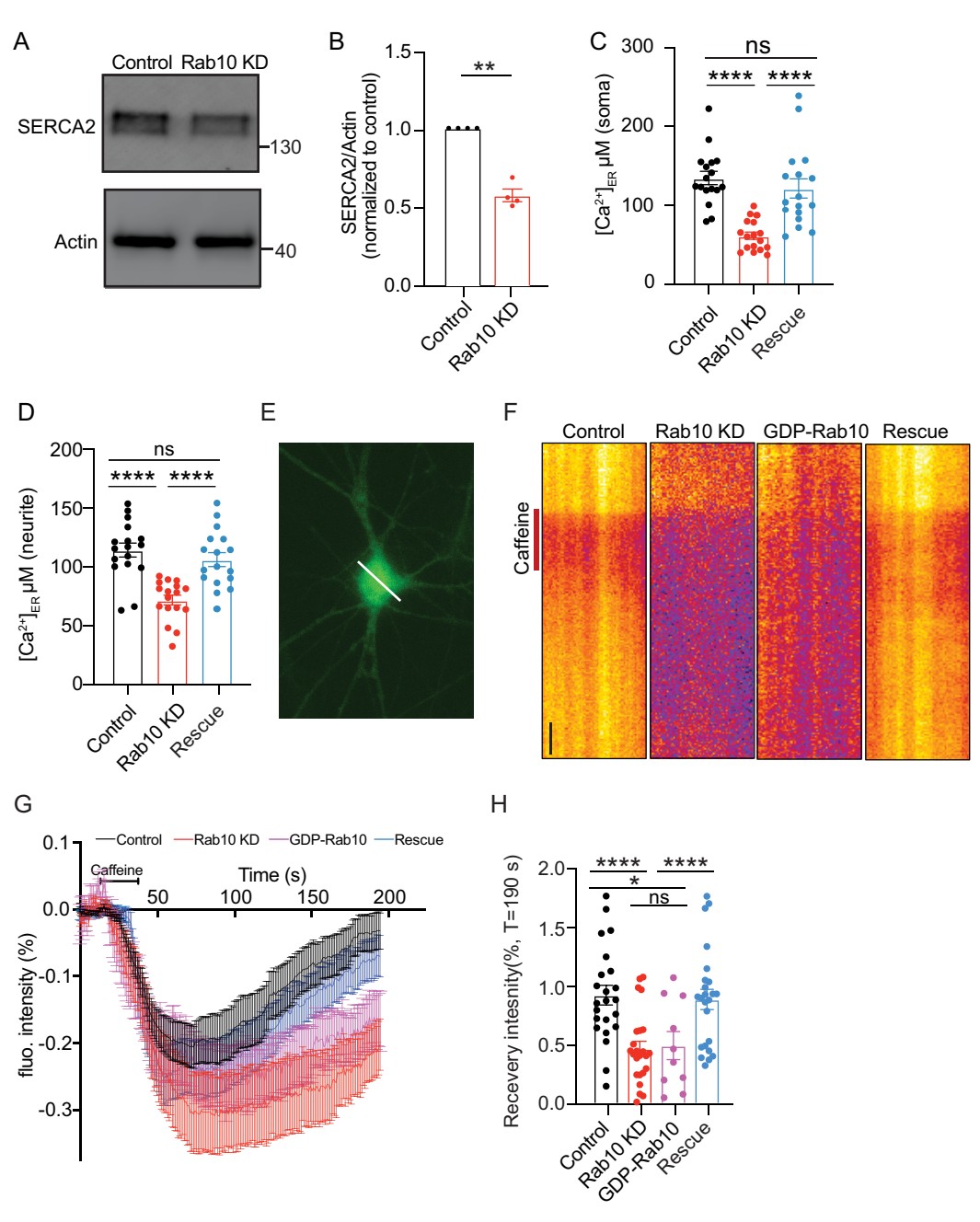

**Figure 5.** Reduced SERCA2 levels and impaired endoplasmic reticulum (ER) Ca$^{2+}$ homeostasis in Rab10 KD neurons. (**A**) Typical immunoblot showing reduced SERCA2 levels in Rab10 KD hippocampal neurons. (**B**) Quantification of protein levels in Rab10 KD neurons normalized to control. (**C**) Quantification of somatic ER Ca$^{2+}$ concentration. (**D**) Quantification of dendritic ER Ca$^{2+}$ concentration. (**E**) Representative image of a neuron infected with ER-GCaMP6-150 displayed with a pseudo line. Scale bar: 3 μm. (**F**) Typical kymographs of the somatic intensity of ER-GCaMP6-150 showing the intensity decrease upon caffeine superfusion (red line) and the recovery in intensity after caffeine washout. Scale bar: 10 s. (**G**) Average normalized ER-GCaMP6-150 fluorescence recovery after caffeine treatment. (**H**) Normalized ER-GCaMP6-150 fluorescence recovery after caffeine treatment at T=190 s. All data are plotted as mean ± s.e.m. (B) Control: N=4, n=4; Rab10 KD: N=4, n=4; (C-D) Control: N=3, n=17; Rab10 KD: N=3; n=17; Rescue: N=3, n=17; (H) Control: N=3, n=23; Rab10 KD: N=3; n=24; GDP-Rab10: n=3, n=10; Rescue: N=3, n=24. A one-way ANOVA tested the significance of adding experimental group as a predictor. ****=p<0.0001, ***=p<0.001, **=p<0.01, ns=not significant.

The online version of this article includes the following source data and figure supplement(s) for figure 5:

*Figure 5 continued on next page*

*Figure 5 continued*

**Source data 1.** PDF file containing original western blots for *Figure 5A*, indicating the relevant bands and treatments.

**Source data 2.** Original files for western blot analysis displayed in *Figure 5A*.

**Figure supplement 1.** Caffeine triggers less endoplasmic reticulum (ER) $Ca^{2+}$ release in Rab10 KD neurons.

*and B*). Therefore, ER alternations in Rab10 KD neurons may disrupt $Ca^{2+}$ homeostasis, which is essential for DCV exocytosis.

To test this, we next measured the $Ca^{2+}$ concentration in ER ($[Ca^{2+}]_{ER}$) using the ER $Ca^{2+}$ indicator ER-GCaMP6 (*de Juan-Sanz et al., 2017*). We observed a reduction in $[Ca^{2+}]_{ER}$ in soma from 130 μM in control neurons to about 70 μM in Rab10-depleted neurons (*Figure 5C*). $[Ca^{2+}]_{ER}$ in neuritis was also reduced by 15% in Rab10 KD neurons (*Figure 5D*). The reduction of $[Ca^{2+}]_{ER}$ was rescued by the expression of a knockdown-resistant Rab10 construct. To validate this observation, we assessed ER $Ca^{2+}$ homeostasis indirectly by measuring the effect of caffeine on cytosolic $Ca^{2+}$ concentration. As expected, in the absence of extracellular $Ca^{2+}$, caffeine application (1 μM) triggered an increase in cytosolic $Ca^{2+}$ due to $Ca^{2+}$ release from the ER in both WT and Rab10-depleted neurons. However, the peak and the area of the caffeine-induced $Ca^{2+}$ response curves were both reduced by 30% in Rab10 KD neurons (*Figure 5—figure supplement 1*).

Furthermore, we examined ER $Ca^{2+}$ dynamics following a 10 min caffeine treatment. Caffeine activates the ryanodine receptor (RyR), leading to the depletion of ER $Ca^{2+}$ (*Endo, 1975*; *Fujimoto et al., 1980*). As expected, perfusion with caffeine induced ER $Ca^{2+}$ depletion, followed by recovery toward pre-stimulation levels (*Figure 5F and G*). In WT neurons, $Ca^{2+}$ levels were recovered by 90% at T=190 s. However, the refilling of ER $Ca^{2+}$ was significantly delayed in Rab10 KD neurons or GDP-Rab10 expressing neurons. $Ca^{2+}$ levels were only recovered by 50% at T=190 s (*Figure 5G and H*) in these neurons.

Finally, to investigate the consequence of the ER $Ca^{2+}$ depletion on neuronal $Ca^{2+}$ homeostasis in Rab10 KD neurons, we measured cytosolic $Ca^{2+}$ responses triggered by APs using the $Ca^{2+}$ indicator Fluo5F (*Figure 6A–C*) and the genetically encoded Synaptophysin-GCaMP6 (*Figure 6D–F*). The AP-induced $Ca^{2+}$ responses in the soma, as measured by Fluo5F, were reduced by 40% upon Rab10 depletion (*Figure 6C*). Similarly, the AP-induced $Ca^{2+}$ responses in presynaptic nerve terminals, measured by Synaptophysin-GCaMP6, were also reduced (20%, *Figure 6F*), although this effect was smaller than the effects of Rab10 KD on ER $Ca^{2+}$ levels and caffeine-induced ER $Ca^{2+}$ depletion (*Figure 5*). Taken together, these data suggest that Rab10 knockdown leads to ER $Ca^{2+}$ depletion and impairs neuronal $Ca^{2+}$ homeostasis, which may be attributed to the reduced levels of SERCA2 level and slower ER $Ca^{2+}$ refilling and may contribute to the impaired DCV exocytosis.

## Rab10 depletion impairs ionomycin-induced DCV exocytosis

To determine whether the impaired $Ca^{2+}$ signaling explains the DCV exocytosis deficiency in Rab10 KD neurons, we stimulated DCV exocytosis using the $Ca^{2+}$ ionophore ionomycin. This approach bypasses cellular $Ca^{2+}$ homeostasis and artificially increases the intracellular $Ca^{2+}$ concentration enough to trigger DCV exocytosis (*Persoon et al., 2019*). Unexpectedly, DCV exocytosis was still reduced in Rab10 KD neurons (*Figure 7A and B*). Although the reduction was substantial, 45%, this impairment was not as substantial as observed for AP-induced DCV exocytosis (65%, *Figure 2K*). The total number of DCV was nearly identical between control and Rab10 KD neurons (*Figure 7D*). Thus, although a minor fraction of the DCV exocytosis deficits may be explained by impaired $Ca^{2+}$ signaling (difference between 45% and 65%), other deficits explain most of the DCV exocytosis deficiency in Rab10 KD neurons.

## Rab10 regulates neuronal protein synthesis

Since significant dysregulation of ER markers and ribosomal proteins was observed upon Rab10 depletion, we investigated the effects of Rab10 on protein synthesis using SUnSET to detect nascent peptides formed during puromycin pulse labeling (*Schmidt et al., 2009*). SUnSET analysis revealed that global protein synthesis was reduced by 30% upon Rab10 depletion (*Figure 8A and B*).

Protein synthesis impairments may be rescued by supplementation with leucine, a branched-chain amino acid, that promotes protein synthesis by activating the mTOR pathway (*Ananieva et al., 2016*).

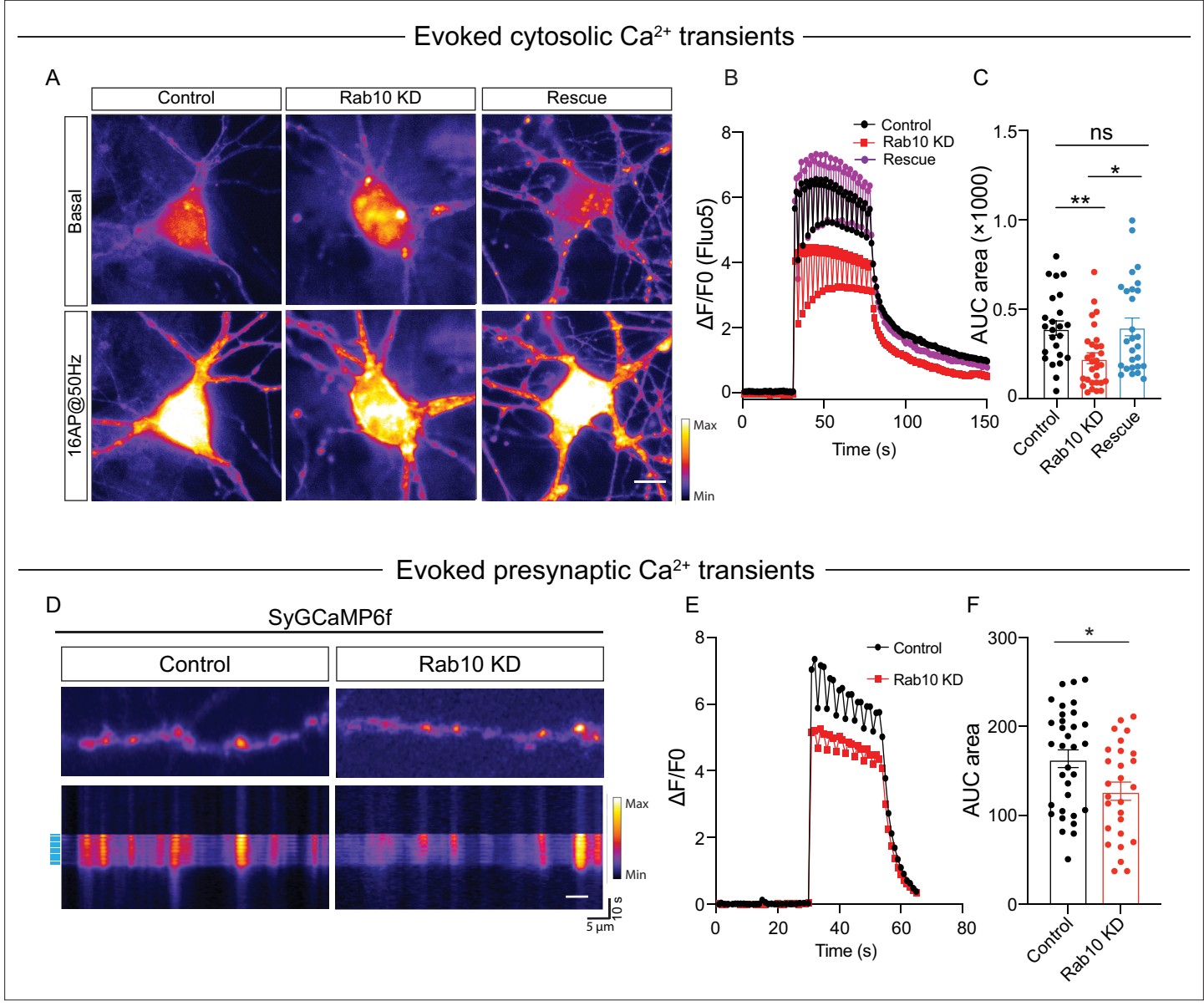

**Figure 6.** Impaired neuronal Ca²⁺ influx triggered by electrical stimulation. (**A**) Representative time-lapse of cytosolic Fluo-5 AM upon electrical stimulation (16 action potentials [APs], 50 Hz) in somas of hippocampal neurons. Scale bar: 10 μm. (**B**) Average normalized response of somatic Fluo-5 AM fluorescence upon stimulation (16 APs, 50 Hz) in hippocampal neurons. (**C**) Quantification of the area under the curve (AUC) of the Fluo-5 AM fluorescence traces. (**D**) Typical neurons infected with Synaptophysin-GCaMP6 (upper), typical kymograph of a neurite (bottom) showing Synaptophysin-GCaMP6 intensity increase upon electrical stimulation (16 APs, 50 Hz, blue bars). Scale bar: 5 μm. (**E**) Average normalized response of Synaptophysin-GCaMP6 fluorescence intensity at presynaptic boutons upon stimulation (16 APs, 50 Hz) in hippocampal neurons. (**F**) Quantification of the AUC of the Synaptophysin-GCaMP6 fluorescence traces in control and Rab10 KD neurons. All data are plotted as mean ± s.e.m. (**C**) Control: N=4, n=24; Rab10 KD: N=4, n=30; Rescue: N=4, n=27. (**F**) Control: N=3, n=33; Rab10 KD: N=3; n=27. A one-way ANOVA tested the significance of adding experimental group as a predictor. **=$p<0.01$, *=$p<0.05$, ns=not significant.

To test this in Rab10 KD neurons, additional L-leucine was added to the culture medium to increase the concentration to 5 mM for 3 days. Indeed, 5 mM leucine supplementation significantly restored global protein synthesis deficits caused by Rab10 depletion (*Figure 8A and B*).

Finally, a similar impairment in global protein synthesis was observed when a loss-of-function mutant of Rab10 (Rab10T23N) was overexpressed in WT neurons (*Figure 8A and B*). The deficit in protein translation is unlikely attributable to the upregulated mTORC1 signaling as the relative phosphorylation level of pS6K1 was unaffected in Rab10 KD neurons (*Figure 8—figure supplement 1*).

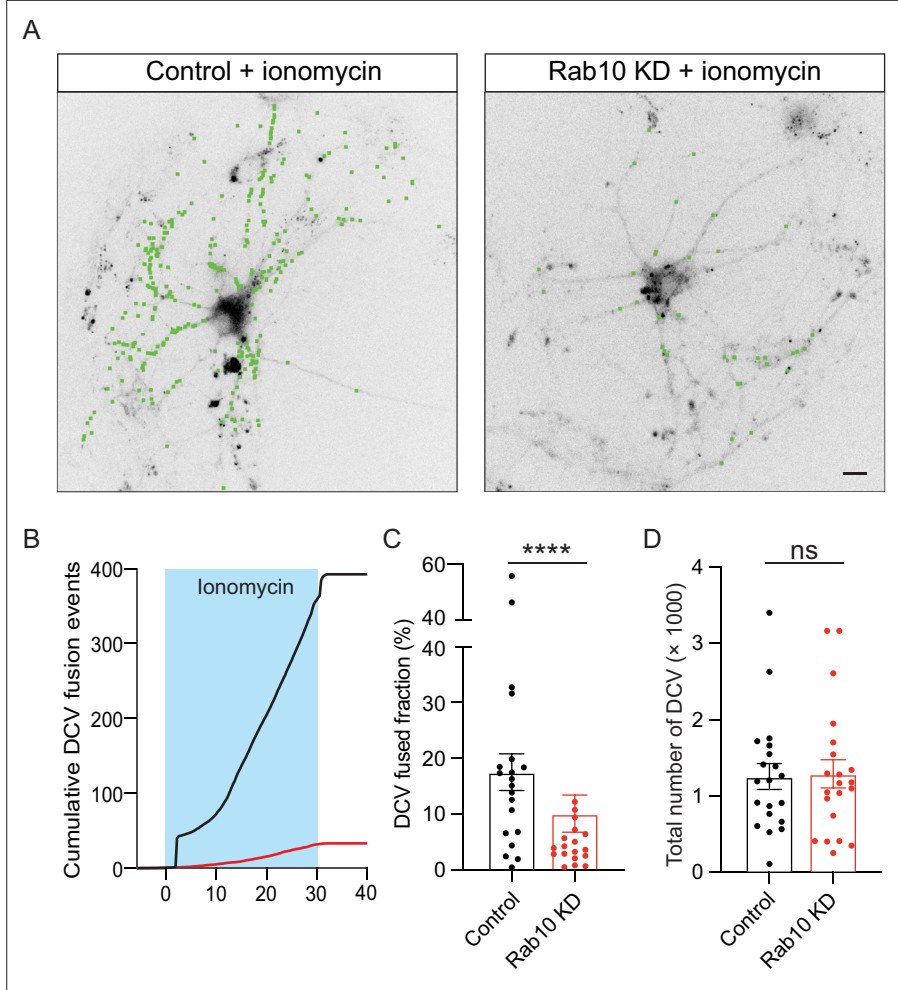

**Figure 7.** Impaired dense core vesicle (DCV) fusion induced by ionomycin in Rab10 KD neurons. (**A**) Representative neurons during electrical stimulation superimposed with NPY- pHluorin fusion events (green dots). Scale bar: 10 μm. (**B**) Cumulative plot of DCV fusion events per cell. (**C**) Fraction of NPY-pHluorin-labeled DCVs fusing during stimulation. (**D**) The total number of DCVs (total pool) of neurons analyzed in B, measured as the number of NPY-pHluorin puncta upon NH$_4$Cl perfusion. All data are plotted as mean ± s.e.m. (C, D) Control: N=3, n=20; Rab10 KD: N=3, n=21. (C, D) A one-way ANOVA tested the significance of adding experimental group as a predictor. *=p<0.05, ns=not significant.

Thus, Rab10 depletion or Rab10T23N expression reduces global protein synthesis in neurons, probably by dysregulation of ER and ribosomal function.

## Leucine supplementation restores normal DCV exocytosis

We hypothesized that protein synthesis deficits in Rab10-depleted neurons explain most of the impaired DCV exocytosis (in addition to a minor fraction explained by disturbed Ca$^{2+}$ homeostasis, see above) and tested whether leucine supplementation could rescue the DCV exocytosis deficits in Rab10 KD neurons. Rab10-depleted neurons expressing NPY-pHluorin were treated with 5 mM leucine 3 days before live-cell imaging or with dimethyl sulfoxide (DMSO) as a control. Leucine supplementation restored DCV exocytosis by 80% caused by Rab10 depletion but did not alter DCV exocytosis in control neurons (*Figure 8C–F*). However, leucine supplementation failed to rescue the defects in ER morphology in Rab10 KD neurons (*Figure 8—figure supplement 2*).

These results suggest that impaired protein synthesis is a major factor contributing to DCV exocytosis deficits in Rab10-depleted neurons.

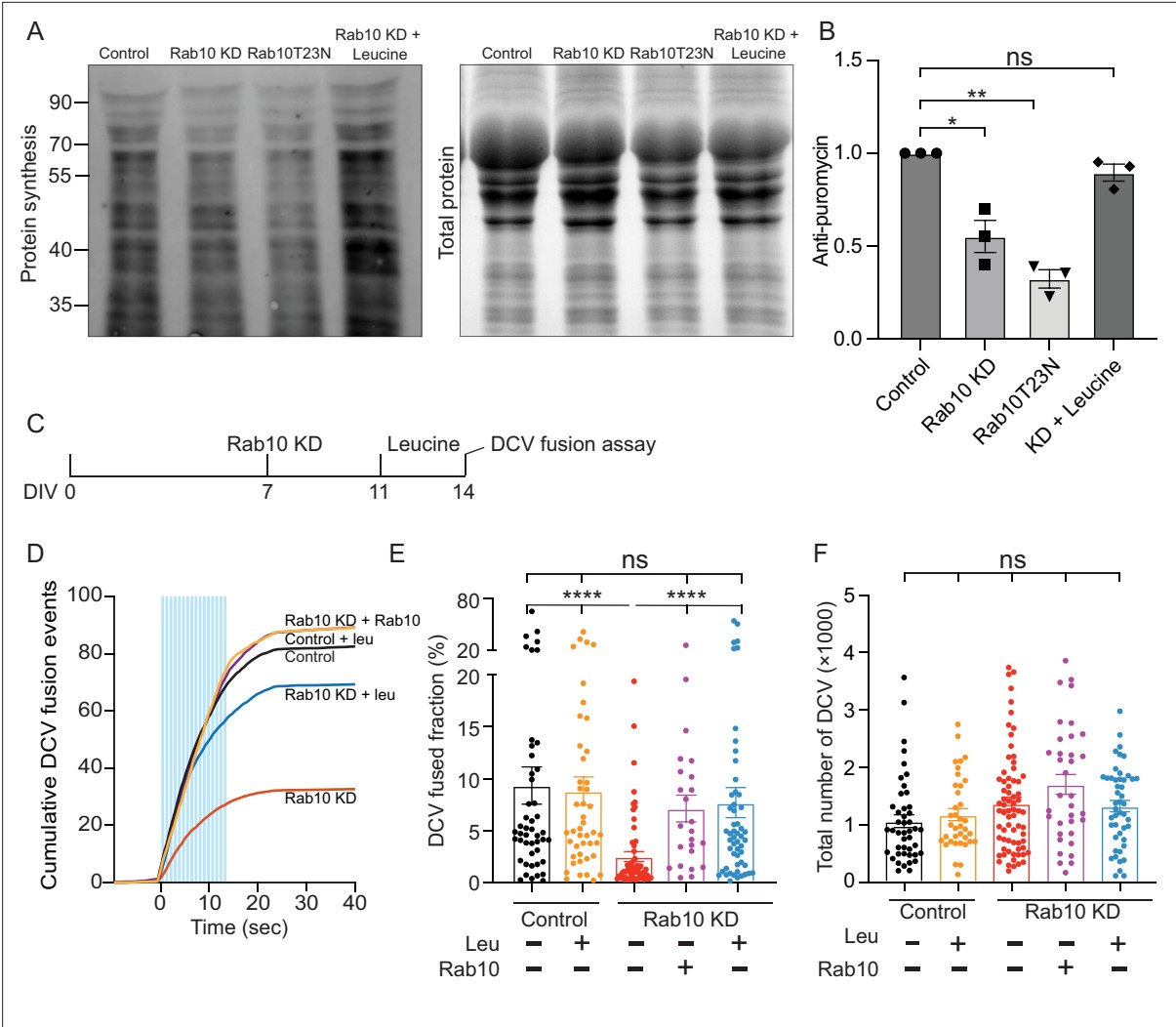

**Figure 8.** Leucine supplementation ameliorates the deficits in protein synthesis and neuropeptide release in Rab10 KD neurons. (**A**) Representative western blot showing puromycinilated proteins as a measure for de novo protein synthesis in each condition. (**B**) Quantification of puromycin intensity in each condition. (**B**) Representation of the dense core vesicle (DCV) fusion assay. Leucine (5 μM) was added to the culture media and incubated for 72 hr before DCV fusion assay. DMSO (1‰) was used as a control. (**C**) Cumulative plot of DCV fusion events per cell. (**D**) Fraction of NPY-pHluorin-labeled DCVs fusing during stimulation. (**E**) The total number of DCVs (total pool) of neurons analyzed in D, E, measured as the number of NPY-pHluorin puncta upon NH₄Cl perfusion. All data are plotted as mean ± s.e.m. (**B**) All: N=3, n=3; (**E, F**) Control: N=3, n=47; Control+leu: N=3, n=45; Rab10 KD: N=3; n=61; Rab10+leu: N=3, n=54. Rab10 KD+Rab10: N=3, n=24. (**B**) One-sample t-test. (**E, F**) A one-way ANOVA tested the significance of adding experimental group as a predictor. **=$p<0.01$, *=$p<0.05$, ns=not significant.

The online version of this article includes the following source data and figure supplement(s) for figure 8:

**Source data 1.** PDF file containing original western blots for *Figure 8A*, indicating the relevant bands and treatments.

**Source data 2.** Original files for western blot analysis displayed in *Figure 8A*.

**Figure supplement 1.** Rab10 depletion does not upregulate mTORC1 pathway.

**Figure supplement 1—source data 1.** PDF file containing original western blots for *Figure 8—figure supplement 1*, indicating the relevant bands and treatments.

**Figure supplement 1—source data 2.** Original files for western blot analysis displayed in *Figure 8—figure supplement 1*.

**Figure supplement 2.** Leucine supplementation does not rescue endoplasmic reticulum (ER) morphological deficiency in Rab10 KD neurons.

**Figure supplement 3.** Overexpression of SERCA2 does not rescue the dense core vesicle (DCV) fusion deficits in Rab10 KD neurons.

## Discussion

In this study, we investigated the function of Rab10 in neuropeptide release in mature mouse hippocampal neurons. We found that DCV exocytosis triggered by AP trains was reduced by 65% upon Rab10 depletion, whereas SV exocytosis was unaffected. In addition, we observed a depleted ER $Ca^{2+}$ pool and an impaired AP-induced $Ca^{2+}$ response in Rab10 KD neurons. However, DCV exocytosis triggered by $Ca^{2+}$ ionophore ionomycin, a triggering method independent of $Ca^{2+}$ channels and internal $Ca^{2+}$ stores, was also impaired, albeit to a lesser extent. Furthermore, ribosomal proteins were massively dysregulated, and protein synthesis was impeded upon Rab10 depletion. Finally, the DCV exocytosis deficit in Rab10 KD neurons was largely rescued by leucine supplementation. We conclude that the strong inhibition of DCV exocytosis upon Rab10 depletion is mostly due to protein synthesis deficiency and to a lesser extent by dysregulation of $Ca^{2+}$ channels or internal $Ca^{2+}$ stores.

### Rab10 regulates neurite outgrowth but not membrane homeostasis in mature neurons

Rab10 is highly enriched in neurons and plays crucial roles in neuronal development (*Taylor et al., 2015*; *Wang et al., 2011*; *Xu et al., 2014*). Consistent with these previous findings, we observed a reduction of axonal and dendritic outgrowth in Rab10-depleted neurons. Also consistent with previous findings in invertebrates (*Sasidharan et al., 2012*), we observed that the endogenous levels of SV markers remain unchanged, indicating that SV biogenesis was unaffected. However, morphological characterization of neurons infected with shRNAs after the first week in culture did not identify changes in the total length of dendrites or axons, indicating that membrane homeostasis in mature neurons was unaffected by Rab10 knockdown. Therefore, the strong deficit in neuropeptide release in DIV7-infected neurons is unlikely to be confounded by Rab10-dependent aspects of neuronal development.

### Rab10 is crucial for DCV exocytosis

In *C. elegans*, neuropeptide release was abolished in Rab10 deletion mutants (*Sasidharan et al., 2012*). We observed a 60% reduction in neuropeptide release in Rab10-depleted mature mouse hippocampal neurons. The difference in effect size could be explained by the incomplete Rab10 depletion with shRNA silencing or by redundant pathways in mammals, e.g., Rab10 and Rab8 are closely related paralogs and share many common effectors (*Homma et al., 2021*).

Unlike Rab3, which travels together DCVs and exhibits a reduction of over 90% in DCV exocytosis in its quadruple knockout neurons (*Persoon et al., 2019*), Rab10 does not typically travel with DCVs (*Figure 2—figure supplement 3*). In addition, no changes in DCV size, puncta intensity, puncta distribution, travel velocity, or distance were detected in Rab10 KD neurons. We conclude that DCV exocytosis deficiency in Rab10-depleted neurons is not caused by alterations in DCV biogenesis or transport and that Rab10 is not required for DCV trafficking.

Although Rab10 is found in subcellular fractions enriched in SVs (*Takamori et al., 2006*; *Taoufiq et al., 2020*), it is dispensable for SV exocytosis. Evoked postsynaptic currents were unaffected in Rab10 mutants in *C. elegans* (*Sasidharan et al., 2012*). In line with this, SV exocytosis was unaffected in Rab10-depleted hippocampal neurons. These results indicate that Rab10 is selectively required for DCV exocytosis, not for SV exocytosis. Strikingly, many synaptic proteins, including many involved in SV exocytosis, are among the most severely dysregulated proteins upon Rab10 depletion. SV exocytosis may be more resilient to acute protein changes (*Sinha et al., 2011*) than DCVs. In addition, vesicle secretion properties are different for DCVs and SVs. Unlike SVs, which are secreted upon a single electrical stimulation, DCVs need prolonged or more intense stimulation for the induction of fusion. Thus, the regulatory effects of Rab10 on DCV exocytosis might be amplified under prolonged stimulation.

### $Ca^{2+}$ homeostasis deficits contribute to DCV exocytosis deficits in Rab10 KD neurons

Rab10 KD neurons showed depleted ER $Ca^{2+}$ and impaired cytosolic $Ca^{2+}$ responses. These effects may contribute to the observed DCV exocytosis deficits. $Ca^{2+}$ released from the ER promotes DCV mobility and potentiates neuropeptide release via activating the CaMKII pathway in *Drosophila* (*Shakiryanova*

*et al., 2007*). However, this might not be the case in mouse neurons since most axonal ER takes $Ca^{2+}$ up from cytosol, instead of releasing it (*de Juan-Sanz et al., 2017*). Second, triggering $Ca^{2+}$ release from the ER did not alter DCV transport and fusion (unpublished data from our lab). Finally, a previous study from our lab has shown that CaMKII deficiency does not alter DCV exocytosis (*Moro et al., 2020*). Hence, dysregulation of ER $Ca^{2+}$ dynamics may not directly explain the observed DCV exocytosis deficits in Rab10-depleted neurons. However, ER $Ca^{2+}$ also regulates $Ca^{2+}$ influx by modulating L-type voltage-gated $Ca^{2+}$ channels at the plasma membrane via an STIM1-based feedback loop (*de Juan-Sanz et al., 2017*). Given the importance of $Ca^{2+}$ influx for DCV exocytosis, dysregulation of ER $Ca^{2+}$ dynamics may indirectly explain DCV exocytosis deficiency upon Rab10 depletion. However, ionomycin-triggered DCV exocytosis, which bypasses voltage-gated $Ca^{2+}$ channels, was still reduced, albeit to a lesser extent compared to AP trains. This difference in effect size is consistent with a limited contribution for dysregulation of $Ca^{2+}$ dynamics and voltage-gated $Ca^{2+}$ channels to explain the impaired DCV exocytosis in Rab10 KD neurons, while the majority of this phenotype is explained by deficits downstream of protein synthesis.

## A role of Rab10 in protein synthesis largely explains DCV exocytosis deficiency in Rab10 KD neurons

In line with the previous study (*Lv et al., 2015*), which indicated alteration in ER morphology in Rab10 KO embryonic cells. We observed impaired ER morphology and upregulated ribosomal proteins upon Rab10 depletion. The upregulation of ribosomal proteins might be a compensatory response to altered ER structure as mutation of other ER-shaping proteins, such as VCP and ALT1, causes similar ribosomal abnormalities as Rab10 depletion (*Shih and Hsueh, 2016*). Protein homeostasis is vital for neuronal function, and deficiency in protein translation is related to several CNS disorders (*Cajigas et al., 2010*; *Holt et al., 2019*; *Koga et al., 2011*; *Laguesse and Ron, 2020*).

Dysregulation of ribosomes and ER is probably sufficient to explain the impaired protein translation in Rab10 KD neurons. Depleted ER $Ca^{2+}$ may induce ER stress (*Arruda and Hotamisligil, 2015*; *Fu et al., 2011*), which may also contribute to protein translation inhibition, but ATF4 levels are unaffected in Rab10 KD neurons, suggesting that ER stress is at best limited (*Figure 4—figure supplement 3A and B*). In addition, Rab10 regulates the retrograde transport of TrkB signaling endosomes (*Lazo and Schiavo, 2023*), which may activate the CREB/mTOR pathway and promote protein synthesis (*Moya-Alvarado et al., 2023*). Hence, impaired TrkB transport may also contribute to impaired protein translation in addition to dysregulated ribosomes in Rab10 KD neurons. We found that restoring protein synthesis with leucine efficiently increased protein synthesis and largely restored DCV exocytosis deficiency in Rab10-depleted neurons. We conclude that dysregulation of protein synthesis results in DCV exocytosis deficits in these neurons.

In conclusion, our data demonstrate the importance of Rab10 in neuropeptide release and ER homeostasis. We observed altered ER morphology, reduced ER $Ca^{2+}$ concentration, impaired protein synthesis, and impaired neuropeptide release in Rab10-depleted neurons. These observations shed light on the pathogenesis of Rab10-related disease. In addition, we have shown that leucine can largely rescue deficiency in protein synthesis and neuropeptide release in Rab10-depleted neurons, providing a potential treatment for disorders associated with neuropeptide abnormalities such as depression and anxiety.

## Materials and methods

### Key resources table

| Reagent type (species) or resource | Designation | Source or reference | Identifiers | Additional information |
|---|---|---|---|---|
| Gene (*Mus musculus*) | *Rab10* | NCBI | 74173 | |
| Genetic reagent (*Mus musculus*) | C57BL/6J | Charles River | Strain code 631 | |
| Genetic reagent (*Rattus norvegicus*) | Wistar (Crl:WI) | Charles River | Strain code 003 | |

*Continued on next page*

*Continued*

| Reagent type (species) or resource | Designation | Source or reference | Identifiers | Additional information |
|---|---|---|---|---|
| Antibody | MAP2 (chicken polyclonal) | Abcam | ab5392 RRID:AB_2138153 | 1:200 (IF) |
| Antibody | SMI312 (mouse polyclonal) | Eurogentec | SMI-312P-050 | 1:500 (IF) |
| Antibody | Synaptophysin 1 (guinea pig polyclonal) | Synaptic Systems | 101004 RRID:AB_1210382 | 1:500 (IF) |
| Antibody | KDEL (mouse monoclonal) | Enzo Life Sciences | ADI-SPA-827-D RRID:AB_2039327 | 1:200 (IF) |
| Antibody | RTN4 (rabbit polyclonal) | Novus Biologicals | NB100-56681 RRID:AB_838641 | 1:200 (IF) |
| Antibody | SERCA2 (mouse monoclonal) | Santa Cruz | sc-376235 RRID:AB_10989947 | 1:200 (IF) |
| Antibody | Rab10 (rabbit polyclonal) | Protein Tech | 11808-1-AP RRID:AB_2173442 | 1:2000 (WB) |
| Antibody | Rab10 (mouse monoclonal) | Abcam | Ab104859 RRID:AB_10711207 | 1:2000 (WB) |
| Antibody | Actin (mouse monoclonal) | Chemicon | MAB1501 RRID:AB_2223041 | 1:4000 (WB) |
| Antibody | Puromycin (mouse monoclonal) | Bio Connect | MABE343 RRID:AB_2566826 | 1:2500 (WB) |
| Antibody | Phospho-p70 S6 Kinase (rabbit monoclonal) | Cell Signaling Technology | 9234S RRID:AB_2269803 | 1:1000 (WB) |
| Antibody | p70 S6 kinase (rabbit polyclonal) | Cell Signaling Technology | 9202S RRID:AB_331676 | 1:1000 (WB) |
| Transfected construct (*Mus musculus*) | shRNA#9 | This paper | – | Lentiviral construct to transfect and express the shRNA (see Materials and methods) |
| Transfected construct (*Mus musculus*) | shRNA#11 | This paper | – | Lentiviral construct to transfect and express the shRNA (see Materials and methods) |
| Transfected construct (*Mus musculus*) | Control | This paper | – | Lentiviral construct to transfect and express the control (see Materials and methods) |
| Recombinant DNA reagent | pLenti-Syn(pr)-NPY-pHluorin | PMID:31679900 | – | – |
| Recombinant DNA reagent | pLenti-Syn(pr)-NPY-mCherry | PMID:31679900 | – | – |
| Recombinant DNA reagent | pLenti-Syn(pr)-Synaptophysin-pHluorin | PMID:34020952 | – | – |
| Recombinant DNA reagent | pLenti-Syn(pr)-Synaptophysin-GCaMP6 | This paper | – | Generation of this reagent is described in Materials and methods |
| Recombinant DNA reagent | ER-GCaMP6-150 | Addgene | RRID:Addgene_86918 | – |
| Recombinant DNA reagent | mCherry-ER3 | Addgene | RRID:Addgene_55041 | – |
| Recombinant DNA reagent | EGFP-Rab10T23N | Addgene | RRID:Addgene_86918 | – |
| Peptide, recombinant protein | pLenti-Syn(pr)- Rab10-EGFP | This paper | – | Generation of this reagent is described in Materials and methods |

*Continued on next page*

*Continued*

| Reagent type (species) or resource | Designation | Source or reference | Identifiers | Additional information |
|---|---|---|---|---|
| Peptide, recombinant protein | 2.5% trypsin | Gibco | 15090046 | – |
| Peptide, recombinant protein | Poly-L-ornithine | Worthington Biochemical Corporation | LS003127 | – |
| Peptide, recombinant protein | Laminin | Sigma-Aldrich | L2020 | – |
| Peptide, recombinant protein | Poly-D-lysine | Sigma-Aldrich | P6407 | – |
| Peptide, recombinant protein | L-Leucine | Sigma-Aldrich | L8000 | – |
| Peptide, recombinant protein | Tunicamycin | Sigma-Aldrich | T7765-10MG | – |
| Chemical compound, drug | Puromycin | Merck/Millipore | 540222-25MG | – |
| Chemical compound, drug | Ionomycin | Fisher Emergo | 10429883 | – |
| Chemical compound, drug | TCE | Sigma-Aldrich | 115-20-8 | – |
| Software, algorithm | MATLAB | MathWorks | RRID:SCR_001622 | – |
| Software, algorithm | Prism | GraphPad | RRID:SCR_002798 | – |
| Other | Fiji/ImageJ | NIH | RRID:SCR_002285 | – |

WB: western blot; IF: immunofluorescence.

## Laboratory animals and primary cultures

All animals were bred and housed according to Institutional and Dutch governmental guidelines and regulations. The primary neuronal culture was done as described before (*Moro et al., 2021*; *Persoon et al., 2019*). Briefly, hippocampi or cortices were extracted from E18 WT embryos in Hanks' Balanced Salt Solution (Sigma-Aldrich), supplemented with 10 mM HEPES (Gibco) and were digested with 0.25% trypsin (Gibco) for 20 min at 37°C. Neurons were washed three times and dissociated with fire-polished Pasteur pipettes. Dissociated neurons were spun down at 1000 rpm for 5 min and resuspended in Neurobasal Medium (Gibco) supplemented with 2% B-27 (Gibco), 1.8% HEPES, 0.25% GlutaMAX (Gibco), and 0.1% penicillin/streptomycin. For continental culture, hippocampal neurons were plated at a density of 30,000 on pre-grown rat glia cells, generated by adding 25,000 glia cells on 18 mm glass coverslips coated with 0.1 mg/ml poly-d-lysine (Sigma-Aldrich) in 12-well plates. For island culture, a density of 1500 hippocampal neurons was plated on pre-grown microglia islands, generated by plating 6000 glia cells on 18 mm glass coverslips coated with agarose and stamped with a solution of 0.1 mg/ml poly-d-lysine (Sigma-Aldrich) and 0.7 mg/ml rat tail collagen (BD Biosciences). For western blot (WB), cortical neurons were plated at a density of 300,000 on six-well plates coated with a solution of 0.0005% poly-l-ornithine and laminin (2.5 μg/ml) (Sigma-Aldrich). Neurons were kept in supplemented Neurobasal at 37°C and 5% $CO_2$ for 14–16 days (DIV14–16).

## Plasmid and lentiviral infection

NPY-pHluorin, NPY-mCherry, and Synaptophysin-pHluorin plasmids have been described (*Persoon et al., 2019*). Synaptophysin-GCaMP6 was generated by adding GCaMP6 to the C-terminus of the mouse sequence of synaptophysin as previously reported (*de Juan-Sanz et al., 2017*). ER-G-CaMP6-150, mCherry-ER3, and Rab10T23N were purchased from Addgene (ER-GCaMP6-150: #86918; mCherry-ER3: #55041; Rab10T23N: #49545). Rab10-EGFP construct was obtained from a mouse cDNA library by PCR and labeled with EGFP at the C-terminus. The target sequences of shRNA are as follows: CGATGCCTTCAATACCACCTT (shRNA#9), GAGAGTTGTACCGAAAGGCAA (shRNA#11), TTC TCCGAACGTGTCACGT (control, scramble), CGATGCATTTAACACAACCTT (Rab10 resistant to shRNA#9).

All plasmids were sequence-verified and packed into lentiviral particles as described previously (*Naldini et al., 1996*).

## Immunocytochemistry

Neurons were fixed at DIV14–16 in freshly prepared 3.7% paraformaldehyde (EMS) for 20 min at room temperature and permeabilized with 0.5% Triton X-100 (Fisher Chemical) for 5 min, blocked with 0.1% Triton X-100 and 2% normal goat serum for 30 min. Incubation with primary antibodies was done at room temperature for 2 hr or overnight at 4°C. After three times of phosphate-buffered saline (PBS) washing, neurons were incubated with Alexa Fluor-conjugated secondary antibodies (1:1000; Invitrogen) for 1 hr at room temperature. Coverslips were mounted in Mowiol (Sigma-Aldrich) and imaged on a Zeiss LSM 510 confocal laser-scanning microscope (×40 objective; NA 1.3) with LSM510 software (version 3.2 Zeiss) or on an A1R Nikon confocal microscope with LU4A laser unit (×40 objective; NA 1.3) with NIS-Elements software (version 4.60, Nikon). Images were acquired as Z-stack at a step of 0.5 µm. All acquisition settings were kept constant for scans within each experiment. All solutions were in PBS (137 mM NaCl, 2.7 mM KCl, 10 mM $Na_2HPO_4$, 1.8 mM $KH_2PO_4$, pH 7.4). Primary antibodies used were: MAP2 (Abcam 1:200), SMI312 (Eurogentec, 1:500), synaptophysin 1 (SySy, 1:500), KDEL (Enzo Life Sciences, 1:200), RTN4 (NB100-56681, 1:200), ATF4 (CST, 1:200). To measure ATF4 intensity, neurons were treated with 5 µg/ml tunicamycin (TM; Sigma-Aldrich) or DMSO as vehicle for 24 hr before fixation. Analysis of staining intensity was done with ImageJ. Analysis of neuronal morphology, synapses, or DCV number was performed using the custom-made software SynD (*Schmitz et al., 2011*).

## Western blotting

Cortical neurons were lysed at DIV14. Lysates were run on a 10% SDS-PAGE gel and transferred to a polyvinylidene difluoride membrane (Bio-Rad). Membranes were blocked with 5% milk (Merck) in PBS with 0.1% Tween 20 for 1 hr at room temperature and incubated in primary antibodies overnight at 4°C. Secondary alkaline phosphatase-conjugated antibodies (1:10,000; Jackson ImmunoResearch) were incubated for 50 min at room temperature. Membranes were visualized with AttoPhos (Promega) and scanned with an FLA-5000 fluorescent image analyzer (Fujifilm). Band intensities of interests were analyzed using Fiji and normalized to the intensity of a loading control (actin).

For de novo-synthesized proteins quantification, surface sensing of translation (SUnSET) was performed as previously described (*Schmidt et al., 2009*). In brief, neurons were incubated with 2 µM puromycin (InvivoGen) for 30 min before harvesting lysates. Puromycinylated proteins were detected with the anti-puromycin antibody by WB. To measure the total protein level, 2,2,2-trichloroethanol (TCE, Lot # BCBK5461V, Sigma-Aldrich) was dissolved in the gel buffer (0.5%) and gels were scanned with Gel Doc EZ Imager (Bio-Rad).

Antibodies used for WB: actin (1:4000; Chemicon), SERCA2 (Santa Cruz, 1:1000), Rab10 (Proteintech, 1:2000), Rab10 (Abcam, 1:2000), Puromycin (Bio Connect, 1:2500), Phospho-p70 S6 kinase (Cell Signaling Technology, 1:1000), p70 S6 kinase (Cell Signaling Technology, 1:1000).

## Proteomics

DIV14 cortical neurons were prepared as previously described (*Gonzalez-Lozano et al., 2019*). In brief, neurons were washed three times with ice-cold PBS. Then, 500 µl PBS supplemented with a protease inhibitor cocktail (Roche) was added to each well and neurons were collected by gentle scraping. Neurons were centrifuged for 5 min at 3000×*g* at 4°C and the pellet was collected and lysed in Laemmli Loading Buffer (4% SDS, 100 mM Tris pH 6.8, 200 mM DTT, 20% glycerol, 0.04% bromophenol blue). In-gel digestion was performed overnight at 37°C with MS grade endo Trypsin/ LysC (Promega). The digested peptides were dried using a SpeedVac and stored at − 20°C until further processing. An SDS-PAGE LC-MS/MS approach was used for peptide identification as previously reported. SWATH data were analyzed using Spectronaut 8.0. The spectral library was created from the merging of two data-dependent analyses of non-transfected hippocampal neuron culture and hippocampal synaptosomes containing spike-in iRT peptides from Biognosys. The retention time prediction was set to dynamic iRT; the cross-run normalization based on total peak areas was enabled. Peptide abundances were exported and analyzed using R language for statistical computation. Only peptides present in both control and transfected groups and quantified with high confidence were

included (i.e. q-value≤10⁻³ over all samples in either group, allowing for one outlier within each condition). Protein abundances were computed using Spectronaut normalized peak area, and Loess normalized using the 'normalizeCyclicLoess' function from the limma R package (fast method and 10 iterations). Proteins with an adjusted FDR≤0.01 and log2 fold change≥0.56 were defined as significant hints. The proteomics experiment presented in *Figure 3* was conducted with two independent cultures with four technical replicates for each condition. For the analysis, we only included peptides that were consistently detected across all samples.

## Bioinformatics

GO analysis on proteomics data was performed with Cytoscope plug-in ClueGO (*Bindea et al., 2009*). The following settings were used for the biological process analysis in ClueGO: Biological process (update: May 25, 2022), GO term grouping, GO tree interval was set 6–10, GO term consists of min. 3 genes and min. 3% of the term. The GO fusion option was set as true with a threshold of 50%. GO terms were grouped with a Kappa score threshold of 0.4 and named after the most significant GO term. Cellular component analysis: Cellular component analysis (update: May 25, 2022), GO term grouping, GO tree interval was set 6–8, GO term consists of min. 5 genes and min. 5% of the term. The GO fusion option was set as true with a threshold of 50%. GO terms were grouped with a Kappa score threshold of 0.5 and named after the most significant GO term. All detected proteins were input as background. GO analysis of synaptic proteins was done with SynGO as previously described (*Koopmans et al., 2019*).

## Electron microscopy

Hippocampal neurons plated on coated plates were infected with control or shRNA#9 at DIV7 and fixed at DIV14 with 2.5% glutaraldehyde in 0.1 M cacodylate buffer (pH 7.4). Samples were post-fixed for 1 hr at room temperature in 1% osmium/1% ruthenium. After dehydration by increasing ethanol concentrations (30%, 50%, 70%, 90%, 96%, and 100%), cells were embedded in EPON solution and polymerized for 72 hr at 65°C. Glass coverslips were removed by heating the sample with hot water. Regions with a high density of neurons were selected under light microscopy and mounted on pre-polymerized EPON blocks. Ultrathin sections (70–90 nm) were cut parallel to the cell monolayer and collected on single-slot, formvar-coated copper grids, and stained in uranyl acetate and lead citrate (Leica EM AC20). Sections were imaged in a JEOL1010 transmission electron microscope (JEOL) at 60 kV while being blinded for the experimental conditions. Synapses, somas, and DCV-rich areas were photographed by a side-mounted Modera camera (EMSIS GmbH). For all synaptic analyses, only synapses with intact synaptic plasma membranes with a recognizable pre- and postsynaptic density and clear SV membranes were selected. DCV and ER diameters were measured in iTEM software (Olympus) and synapse parameters were quantified in a custom-written software running in MATLAB (MathWorks) while being blinded for the experimental conditions.

## Live-cell imaging

Neurons at DIV14–16 were transferred to an imaging chamber and perfused with Tyrode's solution (2 mM CaCl$_2$, 2.5 mM KCl, 119 mM NaCl, 2 mM MgCl$_2$, 30 mM glucose, 25 mM HEPES; pH 7.4). Imaging was acquired on a custom-build microscope (AxioObserver.Z1, Zeiss) with ×40 oil objective (NA 1.3) and an EM-CCD camera (C9100-02; Hamamatsu, pixel size 200 nm) unless otherwise specified. Electrode field stimulation was applied using a stimulus generator (A-385, World Precision Instruments) controlled by a Master-8 (AMPI) to deliver 1 ms pulses of 30 mA. Experiments were performed at room temperature.

For SypHy experiments, neurons were imaged for 30 s as a baseline and then stimulated with electrical field stimulation for 5 s at 40 Hz. After 90 s, neurons were superfused with modified Tyrode's solution containing NH$_4$Cl (2 mM CaCl$_2$, 2.5 mM KCl, 119 mM NaCl, 2 mM MgCl$_2$, 30 mM glucose, 25 mM HEPES, and 50 mM NH$_4$Cl (pH 7.4)) delivered by gravity flow through a capillary placed above the neurons.

SV fusion analysis was performed as described previously (*Moro et al., 2021*). Briefly, regions of interest (ROIs) consisting of 6×6 pixels were placed on individual synapses identified as increased signals after the $NH_4Cl$ perfusion. Individual traces were analyzed using a custom-made MATLAB (MathWorks) script. Synapses were quantified as active if the maximum ΔF/F0 value upon stimulation was ≥3 * StD(F0). Active synapses were pooled per neuron. SV fusion fraction was calculated as the ΔFstimulation/ΔF $NH_4Cl$.

For DCV fusion experiments, the imaging included 30 s of baseline recording and then stimulated with electrical field stimulation for 16 pulses of 50 AP at 50 Hz. Chemical stimulation of 5 µM ionomycin (Fisher Emergo), dissolved in modified Tyrode's solution, was applied through glass capillaries placed near the neuron by gravity flow. After 90 s, neurons were superfused with modified Tyrode's solution containing $NH_4Cl$. For the leucine rescue experiment, neurons expressing NPY-pHluorin were treated with 5 mM leucine 3 days before live-cell imaging or with DMSO as a control.

DCV fusion events were analyzed as described previously (*Persoon et al., 2019*). Briefly, DCV fusion events were detected by a rapid increase in fluorescence intensity. ROIs consisting of 3×3 pixels were placed on the time-lapse recordings using a custom-made script in Fiji. Resulting traces were evaluated using a custom-made script in MATLAB, and only events with F/F0≥2 SD and a rise time of less than 1 s were recorded. F0 was calculated by averaging the first 10 frames of the time-lapse recording. The total intracellular DCV pool was determined as the number of fluorescent puncta after the superfusion of Tyrode's solution containing 50 mM $NH_4Cl$. The released fraction was calculated by dividing the number of fusion events per neuron by the total intracellular pool of DCVs.

For DCV transport experiments, neurons were imaged at DIV14 in time-lapse recordings (2 Hz) at room temperature. Stacks were divided into 10×10 regions with the Grid function in ImageJ, and transport was measured in five random regions (coordinates generated by random number generation in MATLAB). Kymographs were generated in ImageJ (MultipleKymograph, line width 3) and were analyzed with a deep learning-based software (KymoButler) as previously described (*Jakobs et al., 2019*).

## Ca²⁺ imaging

For cytosolic $Ca^{2+}$ imaging, neurons were incubated with 1 µM Fluo-5F AM (Molecular Probes, F14222; stock in DMSO) for 10 min at 37°C. For data shown in *Figure 6D, E, and F*, neurons were perfused with normal Tyrode's solution and stimulated with the same pattern used for DCV experiments.

For the caffeine-induced $Ca^{2+}$ responses (*Figure 5*), neurons were perfused with Tyrode's solution without $Ca^{2+}$. Fluorescent intensity in soma was measured with ImageJ. Normalized ΔF/F0 data was calculated per neuron after background subtraction.

For synaptic $Ca^{2+}$ imaging, neurons were infected with Synaptophysin-GCaMP6 at DIV8 and imaged at DIV14. Neurons were perfused with normal Tyrode's solution and stimulated with the same pattern used for DCV experiments. 20 neurite-located ROIs (6×6 pixels) and a background ROI were measured per cell. Normalized ΔF/F0 data was calculated per neuron after background subtraction.

For ER $Ca^{2+}$ measurement, neurons were infected with ER-GCAMP6-150 at DIV8 and were imaged at DIV14 at room temperature. As previously described (*de Juan-Sanz et al., 2017*), 500 µM or 50 µM ionomycin was applied to saturate the ER-GCAMP6-150 signal in soma or neurite, respectively. $[Ca^{2+}]_{ER}$ were calculated as follows: $[Ca^{2+}]_{ER} = Kd((Fr/Fmax-1/Rf)/(1-Fr/Fmax)^{1/n})$. Kd is the affinity constant of the indicator (150 µM), Fr is the measured fluorescence at rest, Rf is the dynamic range (45), and n is the Hill coefficient (1.6). Fmax values were not corrected for pH changes.

All $Ca^{2+}$ imaging experiments were performed in an imaging buffer with an epifluorescence microscope (Nikon Eclipse Ti) equipped with a ×40 oil objective. Quantitative analysis and image processing were performed using ImageJ.

**Table 1.** Summary of statistical analyses.

| Figure | Dataset | Groups | n-number* | Statistical test | p-value |
|---|---|---|---|---|---|
| 1 A | Band intensity of Rab10 | Control ShRNA#9 ShRNA#11 Rescue | 4 cultures | One sample t-test (compare to 100%) | $P_{shRNA\#9}$=0.0046 (**) $P_{shRNA\#11}$<0.0001 (****) $P_{rescue}$ = 0.5034 (ns) |
| 1 C | Dendritic length (MAP2) | Control ShRNA#9 | 3 (35) 3 (32) | ANOVA model comparison for nested linear models | $P$=0.0093 (**) |
| 1D | Axonal length (SMI312) | Control ShRNA#9 | 3 (35) 3 (32) | ANOVA model comparison for nested linear models | $P$<0.0001 (****) |
| 1E | Syp1 intensity per synapse per neuron | Control ShRNA#9 | 3 (35) 3 (32) | ANOVA model comparison for nested linear models | $P$=0.4975 (ns) |
| 1 F | Syp1-positive synapse density in MAP2-positive dendrites | Control ShRNA#9 | 3 (35) 3 (32) | ANOVA model comparison for nested linear models | $P$=0.4975 (ns) |
| 1 J | SypHy fused fraction | Control ShRNA#9 | 3 (47) 3 (56) | ANOVA model comparison for nested linear models | $P$=0.9496 (ns) |
| 1 K | Decay content | Control ShRNA#9 | 3 (47) 3 (56) | ANOVA model comparison for nested linear models | $P$=0.2910 (ns) |
| 2B | Dendritic length (MAP2) | Control ShRNA#9 ShRNA#11 | 3 (31) 3 (28) 3 (31) | One-way ANOVA | $P$=0.1818 (ns) |
| | | | | ANOVA model comparison for nested linear models | $p_{Control\ vs\ ShRNA\#9}$=0.9771 (ns); $p_{Control\ vs\ ShRNA\#11}$=0.3004 (ns); $p_{ShRNA\#9\ vs\ ShRNA\#11}$=0.2276 (ns); |
| 2 C | Axonal length (SMI312) | Control ShRNA#9 ShRNA#11 | 3 (31) 3 (28) 3 (31) | One-way ANOVA | $P$=0.0936 (ns) |
| | | | | ANOVA model comparison for nested linear models | $p_{Control\ vs\ ShRNA\#9}$=0.5037 (ns); $p_{Control\ vs\ ShRNA\#11}$=0.5313 (ns); $p_{ShRNA\#9\ vs\ ShRNA\#11}$=0.0823 (ns); |
| 2D | Syp1-positive synapse density in MAP2-positive dendrites | Control ShRNA#9 ShRNA#11 | 3 (31) 3 (28) 3 (31) | One-way ANOVA | $P$=0.2126 (ns) |
| | | | | ANOVA model comparison for nested linear models | $p_{Control\ vs\ ShRNA\#9}$=0.3405 (ns); $p_{Control\ vs\ ShRNA\#11}$=0.9788 (ns); $p_{ShRNA\#9\ vs\ ShRNA\#11}$=0.2503 (ns); |
| 2I | DCV fusion events/neuron | Control ShRNA#9 ShRNA#11 Rescue | 3 (36) 3 (37) 3 (30) 3 (34) | One-way ANOVA | $P$<0.0001 (****) |
| | | | | ANOVA model comparison for nested linear models | $p_{Control\ vs\ ShRNA\#9}$=0.0450 (*); $p_{Control\ vs\ ShRNA\#11}$=0.0105 (**); $p_{ShRNA\#11vs\ Rescue}$=0.0021 (**); $p_{ShRNA\#9\ vs\ Rescue}$=0.0100 (*); |
| 2 J | Total DCV pool/neuron | Control ShRNA#9 ShRNA#11 Rescue | 3 (36) 3 (37) 3 (30) 3 (34) | One-way ANOVA | $P$=0.1014 (ns) |
| | | | | ANOVA model comparison for nested linear models | $p_{Control\ vs\ ShRNA\#9}$=0.7669 (ns); $p_{Control\ vs\ ShRNA\#11}$=0.0584 (ns); $p_{ShRNA\#11vs\ Rescue}$=0.4978 (ns); $p_{ShRNA\#9\ vs\ Rescue}$=0.9969 (ns); |
| 2 K | DCV fusion fraction | Control ShRNA#9 ShRNA#11 Rescue | 3 (36) 3 (37) 3 (30) 3 (34) | One-way ANOVA | $P$<0.0001 (****) |
| | | | | ANOVA model comparison for nested linear models | $p_{Control\ vs\ ShRNA\#9}$=0.0014 (**); $p_{Control\ vs\ ShRNA\#11}$=0.0001 (****); $p_{Control\ vs\ Rescue}$=0.9902 (ns); $p_{ShRNA\#9\ vs\ Rescue}$>0.0048 (**); |
| 2 suppl 1D | DCV fusion events/neuron | Control Rab10 KD Rescue | 3 (26) 3 (47) 3 (22) | One-way ANOVA | $P$<0.0001 (****) |
| | | | | ANOVA model comparison for nested linear models | $p_{Control\ vs\ Rab10\ KD}$ = 0.001 (***); $p_{Control\ vs\ Rescue}$>0.9999 (ns); $p_{Rab10\ KD\ vs\ Rescue}$=0.0008 (***); |
| 2 suppl 1E | Total DCV pool/neuron | Control Rab10 KD Rescue | 3 (26) 3 (47) 3 (22) | One-way ANOVA | $P$=0.0021 |
| | | | | ANOVA model comparison for nested linear models | $p_{Control\ vs\ Rab10\ KD}$ = 0.0098(**); $p_{Control\ vs\ Rescue}$=0.9699 (ns); $p_{Rab10\ KD\ vs\ Rescue}$=0.0138 (*); |
| 2 suppl 1 F | DCV fusion fraction | Control Rab10 KD Rescue | 3 (26) 3 (47) 3 (22) | One-way ANOVA | $P$<0.002 (**) |
| | | | | ANOVA model comparison for nested linear models | $p_{Control\ vs\ Rab10\ KD}$=0.0435 (*); $p_{Control\ vs\ Rescue}$=0.6189 (ns); $p_{Rab10\ KD\ vs\ Rescue}$=0.0031 (**); |
| 2 suppl 2B | DCV transport velocity | Control Rab10 KD | 3 (18) 3 (17) | ANOVA model comparison for nested linear models | $P$=0.8028(ns) |
| 2 suppl 2 C | DCV transport distance | Control Rab10 KD | 3 (18) 3 (17) | ANOVA model comparison for nested linear models | $P$=0.9131 (ns) |
| 2 suppl 2 H | Baseline NPY-phluorin intensity | Control Rab10 KD | 3 (37) 3 (35) | ANOVA model comparison for nested linear models | $P$=0.2734 (ns) |
| 2 suppl 2I | NPY-phluorin fusion intensity | Control Rab10 KD | 3 (37) 3 (35) | ANOVA model comparison for nested linear models | $P$=0.3385 (ns) |
| 4 C | Active zone length | Control Rab10 KD | 3 cultures | Linear mixed model | $P$=0.023 (*) |
| 4D | PSD length | Control Rab10 KD | 3 cultures | Linear mixed model | $P$=0.020 (*) |
| 4E | SV number per synapse | Control Rab10 KD | 3 cultures | Linear mixed model | $P$=0.746 (ns) |

*Table 1 continued on next page*

*Table 1 continued*

| Figure | Dataset | Groups | n-number* | Statistical test | p-value |
|---|---|---|---|---|---|
| 4 F | SV diameter | Control Rab10 KD | 3 cultures | Linear mixed model | $P$=0.612 (ns) |
| 4 G | DCV diameter | Control Rab10 KD | 3 cultures | Linear mixed model | $P$=0.260 (ns) |
| 4I | rER diameter | Control Rab10 KD | 3 cultures | Linear mixed model | $P$<0.001 (***) |
| 4 suppl 1B | RTN4 intensity | Control Rab10 KD | 3 (18) 3 (18) | ANOVA model comparison for nested linear models | $P$<0.0001 (****) |
| 4 suppl 1 C | KDEL intensity | Control Rab10 KD | 3 (18) 3 (18) | ANOVA model comparison for nested linear models | $P$<0.0001 (****) |
| 4 suppl 1D | Relative N/S intensity of RTN4 | Control Rab10 KD | 3 (18) 3 (18) | ANOVA model comparison for nested linear models | $P$=0.01551 (*) |
| 4 suppl 1E | Relative N/S intensity of KDEL | Control Rab10 KD | 3 (18) 3 (18) | ANOVA model comparison for nested linear models | $P$<0.0001 (****) |
| 4 suppl 2 C | Recovery intensity of mCherry-ER3 after photobleaching at T=220 s | Control Rab10 KD | 3 (23) 3 (23) | ANOVA model comparison for nested linear models | $P$<0.0001 (****) |
| 4 suppl 3B | ATF4 intensity | Control Rab10 KD TM | 2 (25) 2 (30) 2 (14) | ANOVA model comparison for nested linear models | $p_{Control\ vs\ Rab10\ KD}$=0.1874 (ns); $p_{Control\ vs\ TM}$<0.0001 (****); $p_{Rab10\ KD\ vs\ TM}$<0.0001 (****); |
| 5B | Band intensity of SERCA2 | Control Rab10 KD | 4 cultures | One sample t-test (compare to 100%) | $P$=0.0017 (**) |
| 5 C | Somatic ER Ca²⁺ | Control Rab10 KD Rescue | 3 (17) 3 (17) 3 (17) | One-way ANOVA | $P$<0.0001 (****) |
| | | | | ANOVA model comparison for nested linear models | $p_{Control\ vs\ Rab10\ KD}$<0.0001 (****); $p_{Control\ vs\ Rescue}$>0.5242 (ns); $p_{Rab10\ KD\ vs\ Rescue}$<0.0001 (****); |
| 5D | Neuritic ER Ca²⁺ | Control Rab10 KD Rescue | 3 (17) 3 (17) 3 (17) | One-way ANOVA | $P$<0.0001 (****) |
| | | | | ANOVA model comparison for nested linear models | $p_{Control\ vs\ Rab10\ KD}$<0.0001 (****); $p_{Control\ vs\ Rescue}$>0.5360 (ns); $p_{Rab10\ KD\ vs\ Rescue}$<0.0001 (****); |
| 5 H | Recovery intensity of Fluo-5 AM | Control Rab10 KD GDP-Rab10 Rescue | 3 (23) 3 (24) 3 (10) 3 (24) | One-way ANOVA | $P$<0.0002 (***) |
| | | | | ANOVA model comparison for nested linear models | $p_{Control\ vs\ Rab10\ KD}$=0.0005 (****); $p_{Control\ vs\ Rescue}$>0.9999 (ns); $p_{Rab10\ KD\ vs\ Rescue}$=0.0013 (****); $p_{Control\ vs\ GDP-Rab10}$=0.0307 (*); |
| 5 suppl 1 C | ER Ca²⁺ release triggered by caffeine (peak) | Control Rab10 KD | 3 (44) 3 (35) | ANOVA model comparison for nested linear models | $P$<0.0001 (****) |
| 5 suppl 1D | ER Ca²⁺ release triggered by caffeine (area) | Control Rab10 KD | 3 (44) 3 (35) | ANOVA model comparison for nested linear models | $P$=0.0025 (**) |
| 6 C | Evoked cytosolic Ca²⁺ influx | Control Rab10 KD Rescue | 3 (24) 3 (30) 3 (27) | ANOVA model comparison for nested linear models | $p_{Control\ vs\ Rab10\ KD}$=0.0062 (**); $p_{Control\ vs\ Rescue}$=0.9891 (ns); $p_{Rab10\ KD\ vs\ Rescue}$=0.0128 (*); |
| 6 F | Evoked presynaptic Ca²⁺ influx | Control Rab10 KD | 3 (33) 3 (27) | ANOVA model comparison for nested linear models | $P$=0.0146 (*) |
| 7 C | Ionomycin-induced DCV fused fraction | Control Rab10 KD | 3 (20) 3 (21) | ANOVA model comparison for nested linear models | $P$=0.0009 (****) |
| 7D | Total DCV pool/neuron | Control Rab10 KD | 3 (20) 3 (21) | ANOVA model comparison for nested linear models | $P$=0.8821 (ns) |
| 8B | Band intensity of puromycin | Control Rab10 KD Rab10T23N KD +Leucine | 3 cultures | One sample t-test (compare to 100%) | $P_{Rab10\ KD}$=0.0354 (*) $P_{Rab10\ T23N}$=0.0053 (**) $p_{KD+Leucine}$=0.1486 (ns) |
| 8E | DCV fused fraction | Control Control +Leu Rab10 KD Rab10+Leu Rab10 KD +Rab10 | 3 (47) 3 (45) 3 (61) 3 (54) 3 (24) | One-way ANOVA | $P$<0.0001 (****) |
| | | | | ANOVA model comparison for nested linear models | $p_{Control\ vs\ Rab10\ KD}$<0.0001 (****); $p_{Control\ +\ Leu\ vs\ Rab10\ KD}$<0.0001 (****); $p_{Rab10\ KD\ vs\ Rab10\ KD\ +\ Rab10}$<0.0001 (****); $p_{Rab10\ KD\ vs\ Rab10\ KD\ +\ Leu}$<0.0001 (****); $p_{control\ vs\ Rab10\ KD\ +\ Leu}$=0.577 (ns) |
| 8 F | Total DCV pool/neuron | Control Control +Leu Rab10 KD Rab10+Leu Rab10 KD +Rab10 | 3 (47) 3 (45) 3 (61) 3 (54) 3 (24) | One-way ANOVA | $P$=0.1035 |
| | | | | ANOVA model comparison for nested linear models | $p_{Control\ vs\ Rab10\ KD}$=0.2484 (ns); $p_{Control\ +\ Leu\ vs\ Rab10\ KD}$>0.9999 (****); $p_{Rab10\ KD\ vs\ Rab10\ KD\ +\ Rab10}$>0.9999 (ns); $p_{Rab10\ KD\ vs\ Rab10\ KD\ +\ Leu}$>0.9999 (ns); $p_{control\ vs\ Rab10\ KD\ +\ Leu}$>0.9999 (ns) |
| 8 suppl 2B | KDEL intensity | Control Rab10 KD Rab10+Leu | 3 (10) 3 (11) 3 (11) | ANOVA model comparison for nested linear models | $p_{Control\ vs\ Rab10\ KD}$<0.0001 (****); $p_{control\ vs\ Rab10\ KD\ +\ Leu}$<0.0001 (****); $p_{Rab10\ KDvs\ Rab10\ KD\ +\ Leu}$=0.9970(ns); |
| 8 suppl 2 C | Relative N/S intensity of KDEL | Control Rab10 KD Rab10+Leu | 3 (10) 3 (11) 3 (11) | ANOVA model comparison for nested linear models | $p_{Control\ vs\ Rab10\ KD}$<0.0001 (****); $p_{control\ vs\ Rab10\ KD\ +\ Leu}$<0.0001 (****); $p_{Rab10\ KDvs\ Rab10\ KD\ +\ Leu}$=0.9293(ns); |
| 8 suppl 3 C | DCV fusion events/neuron | Control Rab10 KD SERCA2 | 2 (10) 2 (13) 2 (15) | ANOVA model comparison for nested linear models | $p_{Control\ vs\ Rab10\ KD}$=0.0084 (**); $p_{Control\ vs\ SERCA2}$ = 0.0095 (**); $p_{rab10\ KD\ vs\ SERCA2}$ = 0.0095 (**); |
| 8 suppl 3D | Total DCV pool/neuron | Control Rab10 KD SERCA2 | 2 (10) 2 (13) 2 (15) | ANOVA model comparison for nested linear models | $p_{Control\ vs\ Rab10\ KD}$=0.9988 (ns); $p_{Control\ vs\ SERCA2}$ = 0.9813 (ns); $p_{rab10\ KD\ vs\ SERCA2}$ = 0.9655 (ns); |

*Table 1 continued on next page*

*Table 1 continued*

| Figure | Dataset | Groups | n-number* | Statistical test | p-value |
|--------|---------|--------|-----------|------------------|---------|
| 8 suppl 3E | DCV fused fraction | Control Rab10 KD SERCA2 | 2 (10) 2 (13) 2 (15) | ANOVA model comparison for nested linear models | $p_{Control\ vs\ Rab10\ KD}$=0.0003 (***); $p_{Control\ vs\ SERCA2}$ = 0.0001 (****); $p_{rab10\ KD\ vs\ SERCA2}$ = 0.9711 (ns); |

## Fluorescence recovery after photobleaching

Neurons were infected with mCherry-ER3 at DIV9 and imaged at DIV14 on a Nikon Ti-E Eclipse inverted microscope controlled by NIS-Elements software. The acquisition was performed with a ×40 oil objective. After acquiring 10 pre-FRAP images (every 8.5 s), an 80-pixel long ROI on the proximal axon was photobleached with maximal laser power (10 iterations). Images were acquired for 300 s. The post-bleaching fluorescence intensity was normalized to the baseline fluorescence (F0), which was defined as the average intensity of 10 frames before the onset of photobleaching.

## Statistics

All data are presented as mean ± s.e.m. Datasets on single neuron measurements consist of several neuronal cultures (N=number of independent cultures). Within each culture, different coverslips are infected with various viruses to create distinct experimental groups, from which multiple observations (n=individual neurons) are taken. To account for the nested nature of our datasets, a fixed linear regression was performed, in which culture was included as a linear predictor. Possible outliers were identified using the ROUT method using GraphPad Prism software and were excluded from the statistical analysis. A fixed linear regression model was then fitted to the data using the lm() function in R. A one-way ANOVA (analysis of variance) was used to assess whether including the experimental group as a second linear predictor (formula = y ~ Group + Culture) statistically improved the fit of a model without group information (formula = y ~ 1 + Culture). Post hoc analysis was performed using emmeans() function with Turkey adjustment when more than two experimental groups were present. Full statistical information, including exact p-values, is provided in *Table 1*.

## Acknowledgements

This work was supported by a European Research Council (ERC) Advanced grant (322966) of the European Union (to MV), COSYN (Comorbidity and Synapse Biology in Clinically Overlapping Psychiatric Disorders, Horizon 2020 Program of the European Union under RIA grant agreement 667301, to MV) and the JPND Neuron Cofund ERA-Net SNAREopathy (to RFT). We thank Rien Dekker for the electron microscopy data, Robbert Zalm and Ingrid Saarloos for cloning and producing viral particles, Joke Wortel for the animal breeding, Lisa Laan and Desiree Schut for glia and primary neuron culture assistance, Jurjen Broeke for technical support, and members of the DCV team for discussions and input. The authors declare no competing interests. Correspondence should be addressed to Matthijs Verhage at matthijs@cncr.vu.nl.

## Additional information

### Funding

| Funder | Grant reference number | Author |
|--------|------------------------|--------|
| European Research Council | 322966 | Matthijs Verhage |
| Horizon 2020 Framework Programme | 667301 | Matthijs Verhage |
| ERA-Net NEURON | JPND Neuron Cofund ERA-Net SNAREopathy | Ruud F Toonen |

The funders had no role in study design, data collection and interpretation, or the decision to submit the work for publication.

## Author contributions
Jian Dong, Conceptualization, Data curation, Formal analysis, Investigation, Visualization, Methodology, Writing – original draft, Writing – review and editing; Miao Chen, Jan RT van Weering, Ka Wan Li, Visualization, Methodology; Natalia Domínguez, Investigation, Methodology, Validation; August B Smit, Methodology; Ruud F Toonen, Matthijs Verhage, Conceptualization, Supervision, Funding acquisition, Investigation, Visualization, Methodology, Writing – original draft, Writing – review and editing

## Author ORCIDs
Jian Dong ⓘ https://orcid.org/0009-0002-5201-0748
Jan RT van Weering ⓘ https://orcid.org/0000-0001-5259-4945
Ka Wan Li ⓘ https://orcid.org/0000-0001-6983-5055
Ruud F Toonen ⓘ https://orcid.org/0000-0002-9900-4233
Matthijs Verhage ⓘ https://orcid.org/0000-0002-6085-7503

## Ethics
All animals' experiments were approved by institutional and Dutch Animal Ethical Committee regulations (DEC-FGA 11-03).

Reviewer #1 (Public review): https://doi.org/10.7554/eLife.94930.3.sa1
Reviewer #2 (Public review): https://doi.org/10.7554/eLife.94930.3.sa2
Reviewer #3 (Public review): https://doi.org/10.7554/eLife.94930.3.sa3
Author response https://doi.org/10.7554/eLife.94930.3.sa4

---

# Additional files

## Supplementary files
MDAR checklist

## Data availability
All data generated or analyzed during this study are included in the manuscript and supporting files; source data files have been provided for Figures 1, 5 and 8.

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
