## [Editor Report · eLife Assessment]

In this revised manuscript, Dong et al. investigate the role of the small Ras-like GTPase Rab10 in the exocytosis of DCVs in mouse hippocampal neurons, showing that Rab10 depletion hinders DCV exocytosis independently of its effects on neurite outgrowth. Upon revising their work, these findings provide **compelling** evidence that Rab10 depletion leads to altered ER morphology, impaired ER-based calcium buffering, and decreased ribosomal protein expression, which collectively contributes to defective DCV secretion. The study comes to the **fundamental** conclusion that Rab10 is critical for DCV release by ensuring ER calcium homeostasis.

---

## [Referee Report · Reviewer #1 (Public review)]

Summary:

Dong et al here have studied the impact of the small Ras-like GTPase Rab10 on the exocytosis of dense core vesicles (DVC), which are important mediators of neuropeptide signaling in brain. They use optical imaging to show that lentiviral depletion of Rab10 in mouse hippocampal neurons in culture independent of the established defects in neurite outgrowth hamper DCV exocytosis. They further demonstrate that such defects are paralleled by changes in ER morphology and defective ER-based calcium buffering as well as reduced ribosomal protein expression in Rab10-depleted neurons. Re-expression of Rab10 or supplementation of exogenous L-leucine to restore defective neuronal protein synthesis rescues impaired DCV secretion. Based on these results they propose that Rab10 regulates DCV release by maintaining ER calcium homeostasis and neuronal protein synthesis.

Strengths:

This work provides interesting and potentially important new insights into the connection between ER function and the regulated secretion of neuropeptides via DCVs. The authors combine advanced optical imaging with light and electron microscopy, biochemistry and proteomics approaches to thoroughly assess the effects of Rab10 knockdown at the cellular level in primary neurons. The proteomic dataset provided may be valuable in facilitating future studies regarding Rab10 function. This work will thus be of interest to neuroscientists and cell biologists.

Weaknesses:

Whether and how the phenotypes of Rab10 reported in this study are linked remains an open question. Likewise, a possible role of Rab10 in exocytosis cannot be excluded at this stage.

Comments on revisions:

My previous questions and concerns have been satisfactorily addressed by the authors.

---

## [Referee Report · Reviewer #2 (Public review)]

Summary:

In this paper, the authors assess the function of Rab10 in dense core vesicle (DCV) exocytosis using RNAi and cultured neurons. The author provides evidence that their knockdown (KD) is effective and provides evidence that DCV is compromised. They also perform proteomic analysis to identify potential pathway that are affected upon KD of Rab10 that may be involved in DCV release. Upon focusing on ER morphology and protein synthesis, the authors conclude that defects in protein synthesis and ER Ca2+ homeostasis contributes to the DVC release defect upon Rab10 KD.

Strengths:

The data related to Rab10's role in DCV release seems to be strong and carried out with rigor. While the paper lacks in vivo evidence that this gene is indeed involved in DCV in a living mammalian organism, I feel the cellular studies have value. The identification of ER defect in Rab10 manipulation is not truly novel but it is a good conformation of studies performed in other systems. The finding that DCV release defect and protein synthesis defect seen upon Rab10 KD can be significantly suppressed by Leucine supplementation is also a strength of this work.

Weaknesses:

The weaknesses mentioned in my previous comments have been addressed through the revision process.

---

## [Referee Report · Reviewer #3 (Public review)]

In this study, Dong and colleagues set to dissect the role of Rab10 small GTPase on the intracellular trafficking and exocytosis of dense core vesicles (DCVs). While the authors have already shown that Rab3 plays a central role in the exocytosis of DVC in mammalian neurons, the roles of several other Rab-members have been identified genetically, but their precise mechanism of action in mammalian neurons remains unclear. In this study, the authors use a carefully designed and thoroughly executed series of experiments, including live-cell imaging, functional calcium-imaging, proteomics, and electron microscopy, to identify that DCV secretion upon Rab10 depletion in adult neurons is primarily a result of dysregulated protein synthesis and, to a lesser extent, disrupted intracellular calcium buffering. Given that the full deletion of Rab10 has deleterious effect on neurons and that Rab10 has a major role in axonal development, the authors cautiously employed the knock-down strategy from 7 DIV, to focus on the functional impact of Rab10 in mature neurons. The experiments in this study were meticulously conducted, incorporating essential controls and thoughtful considerations, ensuring rigorous and comprehensive results that fully support the conclusions.

Comments on revisions:

The authors have addressed all the comments and suggestions raised by reviewers, making this an excellent and timely study.

---

## [Author Response]

The following is the authors’ response to the original reviews.

**Public Reviews:**

**Reviewer #1 (Public Review):**
Summary:Dong et al here have studied the impact of the small Ras-like GTPase Rab10 on the exocytosis of dense core vesicles (DVC), which are important mediators of neuropeptide signaling in the brain. They use optical imaging to show that lentiviral depletion of Rab10 in mouse hippocampal neurons in culture independent of the established defects in neurite outgrowth hamper DCV exocytosis. They further demonstrate that such defects are paralleled by changes in ER morphology and defective ER-based calcium buffering as well as reduced ribosomal protein expression in Rab10-depleted neurons. Re-expression of Rab10 or supplementation of exogenous L-leucine to restore defective neuronal protein synthesis rescues impaired DCV secretion. Based on these results they propose that Rab10 regulates DCV release by maintaining ER calcium homeostasis and neuronal protein synthesis.Strengths:This work provides interesting and potentially important new insights into the connection between ER function and the regulated secretion of neuropeptides via DCVs. The authors combine advanced optical imaging with light and electron microscopy, biochemistry, and proteomics approaches to thoroughly assess the effects of Rab10 knockdown at the cellular level in primary neurons. The proteomic dataset provided may be valuable in facilitating future studies regarding Rab10 function. This work will thus be of interest to neuroscientists and cell biologists.

We appreciate the positive evaluation of our manuscript.

Weaknesses:While the main conclusions of this study are comparably well supported by the data, I see three major weaknesses:(1) For some of the data the statistical basis for analysis remains unclear. I.e. is the statistical assessment based on N = number of experiments or n = number of synapses, images, fields of view etc.? As the latter cannot be considered independent biological replicates, they should not form the basis of statistical testing.

This is an important point and we agree that multiple samples from the same biological replicate are not independent observations. We reanalyzed all nested data using a linear mixed model and indicated this in the Methods section and the relevant figure legends (Brunner et al., 2022). In brief, biological replicates (individual neuronal cultures) were used as a linear predictor. Outliers were identified and excluded using the ROUT method in GraphPad. A fixed linear regression model was then fitted to the data using the lm() function in R. A one-way anova (analysis of variance) was used to assess whether including the experimental group as a second linear predictor (formula = y ~ Group + Culture) statistically improved the fit of a model without group information (formula = y ~ 1 + Culture). Post-hoc analysis was performed using the emmeans() function with Tukey’s adjustment when more than two experimental groups were present. Importantly, our conclusions remain unchanged.

(2) As it stands the paper reports on three partially independent phenotypic observations, the causal interrelationship of which remains unclear. Based on prior studies (e.g. Mercan et al 2013 Mol Cell Biol; Graves et al JBC 1997) it is conceivable that defective ER-based calcium signaling and the observed reduction in protein synthesis are causally related. For example, ER calcium release is known to promote pS6K1 phosphorylation, a major upstream regulator of protein synthesis and ribosome biogenesis. Conversely, L-leucine supplementation is known to trigger calcium release from ER stores via IP3Rs. Given the reported impact of Rab10 on axonal transport of autophagosomes and, possibly, lysosomes via JIP3/4 or other mediators (see e.g. Cason and Holzbaur JCB 2023) and the fact that mTORC1, the alleged target of leucine supplementation, is located on lysosomes, which in turn form membrane contacts with the ER, it seems worth analyzing whether the various phenotypes observed are linked at the level of mTORC1 signaling.

This is great suggestion that could indeed further clarify the potential interplay between ER-based Ca2+ signaling and protein synthesis. To address this, we assessed the phosphorylation level of pS6K1 in control and Rab10 knockdown (KD) neurons with or without leucine treatment. These data are included in the new Figure 8—figure supplement 1 in the revised manuscript. Our results indicate that pS6K1 phosphorylation was not upregulated in Rab10 KD neurons, suggesting that the level of mTORC1 signaling is not different between wild-type or KD neurons. Furthermore, leucine treatment increased the pS6K1 phosphorylation level, as expected, but this effect was similar in both groups. Hence, we conclude that differences in mTORC1 signaling induced by Rab10 loss is not a major factor in the observed impairment in protein synthesis.Author response image 1

**Author response image 1. sa4fig1:** Rab10 depletion does not upregulate mTORC1 pathway. (A)Typical immunoblot showing pS6K1 levels in each condition. (B) Quantification of relative pS6K1 levels in each condition. All Data are plotted as mean± s.e.m. (C) Control, Control + Leu: N = 2, n = 2, Rab10 KD, Rab10 KD + Leu: N = 2, n = 4.

(3) The claimed lack of effect of Rab10 depletion on SV exocytosis is solely based on very strong train stimulation with 200 Aps, a condition not very well suited to analyze defects in SV fusion. The conclusion that Rab10 loss does not impact SV fusion thus seems premature.

We agree that 200 APs stimulation might be too strong to detect specific effects on evoked synaptic vesicle release, although this stimulation pattern is an established pattern in hundreds of studies (Emperador-Melero et al., 2018; Granseth et al., 2006; Ivanova et al., 2021; Kwon and Chapman, 2011; Reshetniak et al., 2020). We have toned down our conclusions and clarified in the revised manuscript that Rab10 is dispensable for SV exocytosis evoked by intense stimulations. The corresponding statements in the text have been modified accordingly (p. 5, l. 98, 124) and in figure legend (p. 17, 490).

**Reviewer #2 (Public Review):**
Summary:In this paper, the authors assess the function of Rab10 in dense core vesicle (DCV) exocytosis using RNAi and cultured neurons. The author provides evidence that their knockdown (KD) is effective and provides evidence that DCV is compromised. They also perform proteomic analysis to identify potential pathways that are affected upon KD of Rab10 that may be involved in DCV release. Upon focusing on ER morphology and protein synthesis, the authors conclude that defects in protein synthesis and ER Ca2+ homeostasis contributes to the DVC release defect upon Rab10 KD. The authors claim that Rab10 is not involved in synaptic vesicle (SV) release and membrane homeostasis in mature neurons.Strengths:The data related to Rab10's role in DCV release seems to be strong and carried out with rigor. While the paper lacks in vivo evidence that this gene is indeed involved in DCV in a living mammalian organism, I feel the cellular studies have value. The identification of ER defect in Rab10 manipulation is not truly novel but it is a good conformation of studies performed in other systems. The finding that DCV release defect and protein synthesis defect seen upon Rab10 KD can be significantly suppressed by Leucine supplementation is also a strength of this work.

We appreciate the positive evaluation of our manuscript.

Weaknesses:The data showing Rab10 is NOT involved in SV exocytosis seems a bit weak to me. Since the proteomic analysis revealed so many proteins that are involved in SV exo/encodytosis to be affected upon Rab10, it is a bit strange that they didn't see an obvious defect. Perhaps this could have been because of the protocol that the authors used to trigger SV release (I am not an E-phys expert but perhaps this could have been a 'sledge-hammer' manipulation that may mask any subtle defects)? Perhaps the authors can claim that DCV is more sensitive to Rab10 KD than SV, but I am not sure whether the authors should make a strong claim about Rab10 not being important for SV exocytosis.

We agree that 200 APs stimulation might be too strong to see specific effects on evoked synaptic vesicle release, although this stimulation pattern is an established pattern in hundreds of studies. We have toned down our conclusions and clarified in the revised manuscript that Rab10 is dispensable for SV exocytosis evoked by intense stimulations. The corresponding statements in the text have been modified accordingly (p. 5, l. 98, 124) and in figure legend (p. 17, 490).

Also, the authors mention "Rab10 does not regulate membrane homeostasis in mature neurons" but I feel this is an overstatement. Since the authors only performed KD experiments, not knock-out (KO) experiments, I believe they should not make any conclusion about it not being required, especially since there is some level of Rab10 present in their cells. If they want to make these claims, I believe the authors will need to perform conditional KO experiments, which are not performed in this study.

This is a valid point. We have changed the statement to “membrane homeostasis in mature neurons was unaffected by Rab10 knockdown” (p. 13, l.376-377).

Finally, the authors show that protein synthesis and ER Ca2+ defects seem to contribute to the defect but they do not discuss the relationship between the two defects. If the authors treat the Rab10 KD cells with both ionomycin and Leucine, do they get a full rescue? Or is one defect upstream of the other (e.g. can they see rescue of ER morphology upon Leucine treatment)? While this is not critical for the conclusions of the paper, several additional experiments could be performed to clarify their model, especially considering there is no clear model that explains how Rab10, protein synthesis, ER homeostasis, and Ca2+ are related to DCV (but not SV) exocytosis.

This is an important point and a great suggestion. We have now tested the rescue effects of leucine treatment on ER morphology, as suggested. These data are included in the new Figure 8—figure supplement 2 in the revised manuscript. Our results indicate that the same dose of leucine that rescues DCV fusion and protein translation failed to rescue ER morphology. Hence, the defects in ER morphology appear to be independent of the impaired protein translation.

Author response image 2.

Leucine supplementation does not rescue ER morphological deficiency in Rab10 KD neurons. (A) Typical examples showing the KDEL signals in each condition. (B) Quantification of RTN4 intensity in MAP2-positive dendrites. (C) The ratio of neuritic to somatic RTN4 intensity (N/S).

All Data are plotted as mean ± s.e.m. (B, C) Control: N = 3, n = 10; Rab10 KD: N = 3, n = 11; Rab10 KD + Leu: N = 3; n = 11. A one-way ANOVA tested the significance of adding experimental group as a predictor. **** = p<0.0001, ns = not significant.Author response image 2

**Author response image 2. sa4fig2:** 

**Reviewer #3 (Public Review):**
In the submitted manuscript, Dong and colleagues set out to dissect the role of the Rab10 small GTPase on the intracellular trafficking and exocytosis of dense core vesicles (DCVs). While the authors have already shown that Rab3 plays a central role in the exocytosis of DVC in mammalian neurons, the roles of several other Rab-members have been identified genetically, but their precise mechanism of action in mammalian neurons remains unclear. In this study, the authors use a carefully designed and thoroughly executed series of experiments, including live-cell imaging, functional calcium-imaging, proteomics, and electron microscopy, to identify that DCV secretion upon Rab10 depletion in adult neurons is primarily a result of dysregulated protein synthesis and, to a lesser extent, disrupted intracellular calcium buffering. Given that the full deletion of Rab10 has a deleterious effect on neurons and that Rab10 has a major role in axonal development, the authors cautiously employed the knock-down strategy from 7 DIV, to focus on the functional impact of Rab10 in mature neurons. The experiments in this study were meticulously conducted, incorporating essential controls and thoughtful considerations, ensuring rigorous and comprehensive results.

We are grateful for the positive evaluation of our manuscript.

**Recommendations for the authors:**

**Reviewer #1 (Recommendations For The Authors):**
The work by Dong et al provides interesting and potentially important new insights into the connection between ER function and the regulated secretion of neuropeptides via DCVs. I suggest that the authors address the following points experimentally to increase the impact of this potentially important study.Major points:(1) As alluded to above, for some of the data the statistical basis for analysis remains unclear (examples are Figures 1C-F, J,K; Figure 2 1B-D,I-K; Figure 2 - Supplement 1D-F; Figure 2 - Supplement 2J,K, etc). I.e. is the statistical assessment based on N = number of experiments or n = number of synapses, images, fields of view etc.? As the latter cannot be considered independent biological replicates, they should not form the basis of statistical testing. The Ms misses also misses a dedicated paragraph on statistics in the methods section.

See reply to reviewer 1 above. We fully agree and solved this point.

(2) A main weakness of the paper is the missing connection between neuronal protein synthesis, and the observed structural and signaling defects at the level of the ER. I suggest that the authors analyze mTORC1 signaling in Rab10 depleted neurons and under rescue conditions (+Leu or re-expression of Rab10) as ribosome biogenesis is a major downstream target of mTORC1 and mTORC1 activity is related to lysosome position, which may be affected upon rab10 loss -either directly or via effects on the ER that forms tight contacts with lysosomes.

See reply to reviewer 1 above. We agreed and followed up experimentally.

(3) Related to the above: Does overexpression of SERCA2 restore normal DCV exocytosis in Rab10-depleted neurons? This would help to distinguish whether calcium storage and release at the level of the ER indeed contribute to the exocytosis defect.

This is an important point and a great suggestion. We have now tested the rescue effects of overexpression of SERCA2 on DCV fusion. These data are included in the new Figure 8—figure supplement 3 in the revised manuscript. SERCA2 OE failed to rescue the DCV fusion defects in Rab10 KD neurons.

Author response image 3.

Overexpression of SERCA2 does not rescue DCV fusion deficits in Rab10 KD neurons. (A) Typical examples showing the SERCA2 signals in each condition. (B) Cumulative plot of DCV fusion events per cell. (C) Summary graph of DCV fusion events per cell. (A) Total number of DCVs (total pool) per neuron, measured as the number of NPY-pHluorin puncta upon NH4Cl perfusion. (B) Fraction of NPY-pHluorin-labeled DCVs fusing during stimulation.

All Data are plotted as mean ± s.e.m. (C-E) Control: N = 2, n = 10; Rab10 KD: N = 2, n = 13; SERCA2 OE: N = 2; n = 15. A one-way ANOVA tested the significance of adding experimental group as a predictor. *** = p<0.001, ** = p<0.01, ns = not significant.

\begin{document}$\begin{array}{l} \text { B - Control - Rab10 KD - SERCA2 OE } \\ \text { E } \end{array}$\end{document}

(4) The claimed lack of effect of Rab10 depletion on SV exocytosis is solely based on very strong train stimulation with 200 Aps, a condition not very well suited to analyze defects in SV fusion. The conclusion that Rab10 loss does not impact SV fusion thus seems premature. The authors should conduct additional experiments under conditions of single or few Aps (e.g. 4 or 10 Aps) to really assess whether or not Rab10 depletion alters SV exocytosis at the level of pHluorin analysis in cultured neurons.

See reply to reviewer 2 above. Agreed to and made textual adjustments to solve this

(5) Related to the above: I am puzzled by the data shown in Figure 1H-J: From the pHluorin traces shown I would estimate a tau value of about 20-30 s (e.g. decay to 1/e = 37% of the peak value). The bar graph in Figure 1K claims 3-4 s, clearly clashing with the data shown. Were these experiments conducted at RT (where expected tau values are in the range of 30s) or at 37{degree sign}C (one would expect taus of around 10 s in this case for Syp-pH)? I ask the authors to carefully check and possibly re-analyze their datasets.

This is indeed a mistake. We thank the reviewer for flagging this miscalculation. Our original Matlab script used for calculating the tau value contained an error and the datasets were normalized twice by mistake. We now reanalyzed the data and the corresponding figures and texts have been updated. Our conclusion that Rab10 KD does not affect SV endocytosis remains unchanged since the difference in tau between the control (28.5 s) and Rab10 KD (32.8 s) suffered from the same systematic error and were/are not significantly different.

(6) How many times was the proteomics experiment shown in Figure 3 conducted? I noticed that the data in panel H missed statistical analysis and error bars. Given the typical variation in these experiments, I suggest to only include data for proteins identified in at least 3 out of 4 experimental replicates.

We agree that this information has not been clear. We have now explained replication in the Methods section (p. 42, l. 879-885). In brief, the proteomics experiment presented in Fig 3 was conducted with two independent cultures (‘biological replicates’), hence, formally only two independent observations. For each biological replicate, we performed four technical replicates. For our analysis, we only included peptides that were consistently detected across *all* samples (not only three as this reviewer suggests). Proteins in Panel H are ER-related proteins that are significantly different from control neurons with an adjusted FDR ≤ 0.01 and Log2 fold change ≥ 0.56. The primary purpose of our proteomics experiments was to generate hypotheses and guide subsequent experiments and the main findings were corroborated by other experiments presented in the manuscript.

Minor:(7) Figure 2 - supplement 3 and Figure 4 - supplement 3 are only mentioned in the discussion. The authors should consider referring to these data in the results section.

This is a valid point. We have now added a new statement “Moreover, only 10% of DCVs co-transport with Rab10” in the Results (p. 6-7, l. 162-164).

(8) Where is the pHluorin data shown in Figure 1 bleach-corrected? If so, this should be stated somewhere in the Ms. Moreover, the timing of the NH4Cl pulse should be indicated in the scheme in panel I.

We thank the reviewer for pointing these omissions out. We have now included information about the timing of NH4Cl pulse in panel I. We did not do bleach-correction for the pHluorin data shown in Figure 1. It has been shown that pHluorin is very stable with a bleaching rate in the alkaline state of 0.06% per second and 0.0024% per second in the quenched state (Balaji and Ryan, 2007). Indeed, we did not observe obvious photobleaching in the first 30s during our imaging as indicated by the average trace of pHluorin intensity in panel I.

(9) Page 3/ lines 59-60: "...strongest inhibition of neuropeptide accumulation...". What is probably meant is "...strongest inhibition of neuropeptide release".

We agree this statement is unclear. Sasidharan et al used a coelomocyte uptake assay as an indirect readout for DCV release. The ‘strongest inhibition of neuropeptide accumulation’ in coelomocytes in Rab10 mutant indicates DCV fusion deficits. We have now replaced the text with “Rab10 deficiency produces the strongest inhibition of neuropeptide release in *C. elegans*” to make it more clear.

**Reviewer #3 (Recommendations For The Authors):**
I strongly recommend the publishing of this study as a VOR with minor comments directed to the authors.(1) In Figure 4, the authors should include examples of tubular ER at the synapse, especially as this is an interesting point discussed in ln 226-229. Are there noticeable changes in the ER-mitochondria contacts at the synaptic boutons?

We agree that examples of tubular ER at the synapse would improve the manuscript. We have now replaced the Figure 4A with such examples. We found it challenging to quantify ER-mitochondria contacts based on the electron microscopy (EM) images we currently have. The ER-mitochondria contact sites are quite rare in the cross-sections of our samples, making it difficult to perform a reliable quantitative analysis.

(2) The limited impairment of calcium-ion homeostasis in Rab10 KD neurons is very interesting. Would the overexpression of Rab10T23N mimic the effect of a KD scenario? Is there a separation of function for Rab10 in calcium homeostasis vs. the regulation of protein synthesis?

This is an interesting possibility. We tested this and expressed Rab10T23N in a new series of experiments. These data are presented as a new Figure 5 in the revised manuscript (p. 29). We observed that Ca2+ refilling after caffeine treatment was delayed to a similar extent in Rab10T23N-expressing and Rab10 KD neurons. While impaired Ca2+ homeostasis may affect protein synthesis through ER stress or mTORC1 activation, our findings indicate otherwise in Rab10 KD neurons. First, ATF4 levels, a marker of ER stress, were unaffected in Rab10 KD neurons. This indicates that any ER stress present is minimal or insufficient to significantly impact protein synthesis through this pathway. Second, we did not observe significant changes in mTORC1 activation in Rab10 KD neurons as indicated by a normal pS6K1 phosphorylation (see above). Based on these observations, we conclude that Rab10's roles in calcium homeostasis and protein synthesis are most likely separate.

(3) The authors indicate that the internal release of calcium ions from the ER has no effect on DCV trafficking and fusion without showing the data. It is important to include this data as the major impact of the study is the dissecting of the calcium effects in mammalian neurons from the previous studies in invertebrates.

We agree this is an important aspect in our reasoning. We are submitting the related manuscript on internal calcium stores to BioRVix. The link will be added to the consolidated version of our manuscript

(4) The distinction between Rab3 and Rab10 co-trafficking on DCVs should be reported in the Results (currently, Figure 2 - supplement 3 is only mentioned in the Discussion) as it helps to understand the effects on DCV fusion.

We agree. We now added a new statement “Moreover, only 10% of DCVs co-transport with Rab10” in the Results (p. 6, l. 162-163).

Reference:

Balaji, J., Ryan, T.A., 2007. Single-vesicle imaging reveals that synaptic vesicle exocytosis and endocytosis are coupled by a single stochastic mode. Proceedings of the National Academy of Sciences 104, 20576–20581. https://doi.org/10.1073/pnas.0707574105

Brunner, J.W., Lammertse, H.C.A., Berkel, A.A. van, Koopmans, F., Li, K.W., Smit, A.B., Toonen, R.F., Verhage, M., Sluis, S. van der, 2022. Power and optimal study design in iPSC-based brain disease modelling. Molecular Psychiatry 28, 1545. https://doi.org/10.1038/s41380-022-01866-3

Emperador-Melero, J., Huson, V., van Weering, J., Bollmann, C., Fischer von Mollard, G., Toonen, R.F., Verhage, M., 2018. Vti1a/b regulate synaptic vesicle and dense core vesicle secretion via protein sorting at the Golgi. Nat Commun 9, 3421. https://doi.org/10.1038/s41467-018-05699-z

Granseth, B., Odermatt, B., Royle, S.J., Lagnado, L., 2006. Clathrin-Mediated Endocytosis Is the Dominant Mechanism of Vesicle Retrieval at Hippocampal Synapses. Neuron 51, 773–786. https://doi.org/10.1016/j.neuron.2006.08.029

Ivanova, D., Dobson, K.L., Gajbhiye, A., Davenport, E.C., Hacker, D., Ultanir, S.K., Trost, M., Cousin, M.A., 2021. Control of synaptic vesicle release probability via VAMP4 targeting to endolysosomes. Science Advances 7, eabf3873. https://doi.org/10.1126/sciadv.abf3873

Kwon, S.E., Chapman, E.R., 2011. Synaptophysin Regulates the Kinetics of Synaptic Vesicle Endocytosis in Central Neurons. Neuron 70, 847–854. https://doi.org/10.1016/j.neuron.2011.04.001

Reshetniak, S., Fernández-Busnadiego, R., Müller, M., Rizzoli, S.O., Tetzlaff, C., 2020. Quantitative Synaptic Biology: A Perspective on Techniques, Numbers and Expectations. International Journal of Molecular Sciences 21, 7298. https://doi.org/10.3390/ijms21197298